# Generative Model for Change Point Detection in Dynamic Graphs

## Abstract

This paper proposes a generative model to detect change points in time series of graphs. The proposed framework consists of learnable prior distributions for low-dimensional graph representations and of a decoder that can generate graphs from the latent representations. The prior distributions of the latent spaces are learned from the observed data as empirical Bayes and generative model is employed to assist multiple change point detection. Specifically, the model parameters are learned via maximum approximate likelihood, with a Group Fused Lasso regularization on the prior parameters. The optimization problem is then solved via Alternating Direction Method of Multipliers and Langevin Dynamics are recruited for posterior inference. Simulation studies show good performance of the generative model in supporting change point detection, and real data experiments yield change points that align with significant events.

## 1 Introduction

Networks are often used to represent relational phenomena in numerous domains (Dwivedi et al., 2021; He et al., 2023; Han et al., 2023) and relational phenomena by nature progress in time. In recent decades, a plethora of network models has been proposed to analyze the interaction between objects or people over time, including Temporal Exponential-family Random Graph Model (Hanneke et al., 2010; Krivitsky & Handcock, 2014), Stochastic Actor-Oriented Model (Snijders, 2001; Snijders et al., 2010), and Relational Event Model (Butts, 2008; Butts et al., 2023). Although these models incorporate the temporal aspect for network analysis, network evolution is usually time-heterogeneous. Without taking the structural changes across dynamic networks into consideration, learning from the time series may lead to ambiguity. Hence, it is practical for researchers to localize the change points before studying the evolving networks.

More recently, various methodologies have been proposed to detect change points in dynamic networks. Chen et al. (2020) and Shen et al. (2023) employed embedding methods to detect both anomalous graphs and vertices in time series of networks. Park & Sohn (2020) combined the multi-linear tensor regression model with a hidden Markov model, detecting changes based on the transition between the hidden states. Sulem et al. (2023) learned a graph similarity function using a Siamese graph neural network to differentiate the graphs before and after a change point. Zhao et al. (2019) developed a screening algorithm that is based on an initial graphon estimation to detect change points. Huang et al. (2020) utilized the singular values of the Laplacian matrices as graph embedding to detect the differences across time. Chen & Zhang (2015), Chu & Chen (2019), and Song & Chen (2022a) proposed a non-parametric approach to delineate the distributional differences over time, and Garreau & Arlot (2018) and Song & Chen (2022b) exploited the patterns in high dimensions via a kernel-based method. Zhang et al. (2024) jointly trained a Variational Graph Auto-Encoder and Gaussian Mixture Model to detect the change points.

Inherently, network structures are complex due to highly dyadic dependency. Acquiring a low dimensional representation of a graph can summarize the enormous amount of individual relations to promote downstream analysis (Gallagher et al., 2021). In particular, Sharifnia & Saghaei (2022) and Kei et al. (2023) proposed to detect the structural changes using an Exponential-family Random Graph Model. Yet they relied on user-specified network statistics, which are usually not known to the modeler a priori. Moreover, Larroca et al.

(2021), Marenco et al. (2022), and Gong et al. (2023) developed different latent space models for dynamic graphs to detect changes, but they focused on node level representation, which may not be powerful enough to capture the information of the entire graph. Consequently, we aim to infer the graph level representations that induce the structural changes to facilitate the detection.

On the other hand, generative models recently showed promising results in myriad applications, such as text generation with Large Language Model (Devlin et al., 2018; Lewis et al., 2019) and image generation with Diffusion Model (Ho et al., 2020; Rombach et al., 2022). Similarly, we aim to explore how generative models can assist change point detection in dynamic graphs. In particular, Simonovsky & Komodakis (2018) proposed a Graph Variational Auto-Encoder (VAE) for graph generation, with a zero-mean Gaussian prior to regularize the latent space of the graph level representation. In the VAE framework (Kingma, 2013; Kipf & Welling, 2016), regularization via Kullback Leibler divergence arises from the Evidence Lower Bound for the marginal likelihood, encouraging the approximate posterior to be close to the zero-mean Gaussian prior. In contrast, we focus on learning the mean of the Gaussian prior at each time point and we apply Group Fused Lasso regularization to promote sparsity in the sequential differences of the prior parameters, effectively smoothing out minor fluctuations and highlighting significant change points.

To tackle the challenges in dynamic graphs and to employ recent advances in generative models, we make the following contributions in this manuscript:

- We learn graph level representations of network structures to facilitate change point detection. We impose a prior distribution to the representation at each time point and a multivariate total variation regularization to the sequential differences of the prior parameters. The prior distributions and a graph decoder are jointly learned via maximum approximate likelihood.

- We derive an Alternating Direction Method of Multipliers (ADMM) procedure to solve the optimization problem. Without using an encoder, the prior distributions and the graph decoder are learned by inferring from the posterior distribution via Langevin Dynamics. Experiments show good performance of the generative model in supporting change point detection.

The rest of the manuscript is organized as follows. Section 2 specifies the proposed framework. Section 3 presents the objective function with Group Fused Lasso regularization and the ADMM procedure to solve the optimization problem. Section 4 discusses change points localization and model selection. Section 5 illustrates the proposed method on simulated and real data. Section 6 concludes the work with a discussion and potential future developments.

## 2 Generative Model for Change Point Detection

### 2.1 Model Specification

For a node set $N = \{1, 2, \cdots, n\}$, we use an adjacency matrix $\boldsymbol{y} \in \{0, 1\}^{n \times n}$ to represent a graph. We denote the set of all possible node pairs as $\mathbb{Y} = N \times N$. In the adjacency matrix, $\boldsymbol{y}_{ij} = 1$ indicates an edge between nodes $i$ and $j$, while $\boldsymbol{y}_{ij} = 0$ indicates no edge. The relations can be either directed or undirected. The undirected variant has $\boldsymbol{y}_{ij} = \boldsymbol{y}_{ji}$ for all $(i, j) \in \mathbb{Y}$.

Denote $\boldsymbol{y}^t$ as a network at a discrete time point $t$. The observed data is a sequence of networks $\boldsymbol{y}^1, \ldots, \boldsymbol{y}^T$. For each network $\boldsymbol{y}^t$, we assume there is a latent variable $\boldsymbol{z}^t \in \mathbb{R}^d$ such that the network $\boldsymbol{y}^t$ can be generated from the latent variable with the following graph decoder:

$$\boldsymbol{y}^t \sim P(\boldsymbol{y}^t | \boldsymbol{z}^t) = \prod_{(i,j) \in \mathbb{Y}} \text{Bernoulli}(\boldsymbol{y}^t_{ij}; \boldsymbol{r}_{ij}(\boldsymbol{z}^t))$$

where $\boldsymbol{r}_{ij}(\boldsymbol{z}^t) = P(\boldsymbol{y}^t_{ij} = 1 | \boldsymbol{z}^t)$ is the Bernoulli parameter for dyad $(i, j)$ and it is elaborated in Section 2.2. Conditioning on the latent variable $\boldsymbol{z}^t$, we assume the network $\boldsymbol{y}^t$ is dyadic independent.

We also impose a learnable prior distribution to the latent variable as

$$\boldsymbol{z}^t \sim P(\boldsymbol{z}^t) = \mathcal{N}(\boldsymbol{z}^t; \boldsymbol{\mu}^t, \boldsymbol{I}_d)$$

77 where $\boldsymbol{\mu}^t \in \mathbb{R}^d$ and $\boldsymbol{I}_d$ is an identity matrix. With the graph decoder $P(\boldsymbol{y}^t|\boldsymbol{z}^t)$, we consider $\boldsymbol{z}^t \in \mathbb{R}^d$ as a
78 graph level representation for $\boldsymbol{y}^t \in \{0,1\}^{n \times n}$.

## 2.2 Graph Decoder

The graph decoder $P(\boldsymbol{y}^t|\boldsymbol{z}^t)$ is formulated with a Bernoulli parameter for dyad $(i,j)$ as

$$\boldsymbol{r}_{ij}(\boldsymbol{z}^t) = P(\boldsymbol{y}_{ij}^t = 1|\boldsymbol{z}^t) = \boldsymbol{g}_{ij}\big(\boldsymbol{h}(\boldsymbol{z}^t)\big).$$

80 The function $\boldsymbol{h}(\cdot)$ is parameterized by neural networks with $\boldsymbol{h} : \mathbb{R}^d \to \mathbb{R}^{n \times n}$. The function $\boldsymbol{g}(\cdot)$ is the
81 element-wise sigmoid function with $\boldsymbol{g} : \mathbb{R}^{n \times n} \to [0,1]^{n \times n}$.

In particular, we use multi-layer perceptrons, transferring the latent variable $\boldsymbol{z}^t \in \mathbb{R}^d$ to $\boldsymbol{U}^t \in \mathbb{R}^{n \times k}$ and
$\boldsymbol{V}^t \in \mathbb{R}^{n \times k}$, respectively. We let the latent dimension $d$ and $k$ be smaller than the number of nodes $n$, and

$$\boldsymbol{h}(\boldsymbol{z}^t) = \begin{cases} \boldsymbol{U}^t \boldsymbol{V}^{t\top} \in \mathbb{R}^{n \times n}, & \text{for directed network}, \\ \boldsymbol{U}^t \boldsymbol{U}^{t\top} \in \mathbb{R}^{n \times n}, & \text{for undirected network}. \end{cases}$$

82 The graph decoder via matrix multiplication is common in the literature (Kipf & Welling, 2016; Hamilton
83 et al., 2017; Pan et al., 2018). Comparing to a decoder that directly outputs the graph $\boldsymbol{y}^t \in \{0,1\}^{n \times n}$, the
84 decoder via matrix multiplication reduces the number of parameters in the neural networks and helps avoid
85 over-parameterization, which is crucial for graphs that are sparse.

86 Figure 1 gives an overview of the proposed framework. Implicitly, the graph level representation $\boldsymbol{z}^t$ progresses
87 to node level representations $\boldsymbol{U}^t$ and $\boldsymbol{V}^t$ as an intermediate step, before the generation of network $\boldsymbol{y}^t$. The
88 graph decoder $P_\phi(\boldsymbol{y}^t|\boldsymbol{z}^t)$ with neural network parameter $\boldsymbol{\phi}$ is shared across $t = 1, \ldots, T$. It is worth pointing
89 out the simplicity of our framework, without the need of encoders.

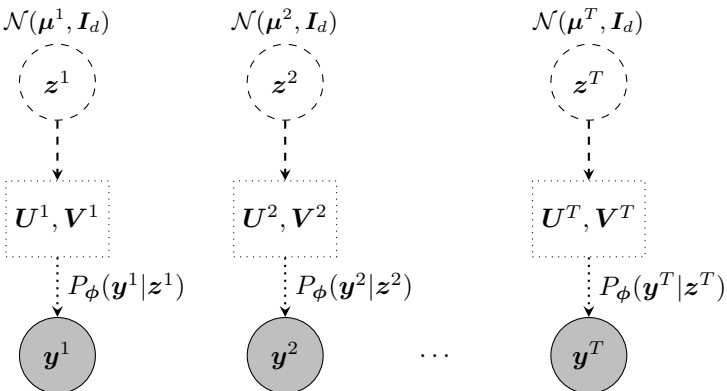

Figure 1: An overview of prior distributions and graph decoder.

## 2.3 Change Points

Anchored on the proposed framework, we can now specify the change points to be detected, in terms of the
prior parameters $\boldsymbol{\mu}^t \in \mathbb{R}^d$ for $t = 1, \ldots, T$. Let $\{C_k\}_{k=0}^{K+1} \subset \{1, 2, \ldots, T\}$ be a collection of ordered change
points with $1 = C_0 < C_1 < \cdots < C_K < C_{K+1} = T$ such that

$$\boldsymbol{\mu}^{C_k} = \boldsymbol{\mu}^{C_k+1} = \cdots = \boldsymbol{\mu}^{C_{k+1}-1}, \ \ k = 0, \ldots, K,$$

$$\boldsymbol{\mu}^{C_k} \neq \boldsymbol{\mu}^{C_{k+1}}, \ \ k = 0, \ldots, K-1, \ \text{ and } \ \boldsymbol{\mu}^{C_{K+1}} = \boldsymbol{\mu}^{C_K}.$$

91 The associated multiple change point detection problem comprises recovering the collection $\{C_k\}_{k=1}^K$ from a
92 sequence of observed networks $\{\boldsymbol{y}^t\}_{t=1}^T$, where the number of change points $K$ is also unknown.

To facilitate change point detection for $\{\boldsymbol{y}^t\}_{t=1}^T$ in the data space, we turn to learn the prior parameters $\{\boldsymbol{\mu}^t\}_{t=1}^T$ in the latent space. Intuitively, the consecutive prior parameters are similar when no change occurs, but they are different when a change emerges. For notational simplicity, we denote $\boldsymbol{\mu} \in \mathbb{R}^{T \times d}$ as a matrix where the $t$-th row corresponds to $\boldsymbol{\mu}^t \in \mathbb{R}^d$ with $t = 1, \ldots, T$.

## 3  Learning and Inference

### 3.1  Learning Priors from Dynamic Graphs

Inspired by Vert & Bleakley (2010) and Bleakley & Vert (2011), we formulate the change point detection problem as a Group Fused Lasso problem (Alaíz et al., 2013). Denote the log-likelihood of the distribution for $\boldsymbol{y}^1, \ldots, \boldsymbol{y}^T$ as $l(\boldsymbol{\phi}, \boldsymbol{\mu})$. We want to solve

$$\hat{\boldsymbol{\phi}}, \hat{\boldsymbol{\mu}} = \operatorname*{arg\,min}_{\boldsymbol{\phi}, \boldsymbol{\mu}} -l(\boldsymbol{\phi}, \boldsymbol{\mu}) + \lambda \sum_{t=1}^{T-1} \|\boldsymbol{\mu}^{t+1} - \boldsymbol{\mu}^t\|_2 \tag{1}$$

where $\lambda > 0$ is a tuning parameter for the Group Fused Lasso penalty term.

The Group Fused Lasso penalty is useful for change point detection because it enforces piecewise constant patterns in the learned parameters by minimizing the total variation. Specifically, the regularization term, expressed as the sum of the $\ell_2$ norms, encourages sparsity of the differences $\boldsymbol{\mu}^{t+1} - \boldsymbol{\mu}^t \in \mathbb{R}^d$, while allowing multiple coordinates across the $d$ dimensional differences to change at the same time $t$. The latter is often referred as a grouping effect that could not be achieved with the $\ell_1$ penalty of the differences. Furthermore, since the regularization is imposed on the prior parameters that relate to the likelihood of the data, the learned priors incorporate the structural changes from the observed graphs into the latent space. In summary, by penalizing the sum of sequential differences, the proposed framework focuses on capturing meaningful structural changes and smoothing out minor variations.

To solve the optimization problem in (1) that involves latent variables, we need to manipulate the objective function accordingly. We first introduce a slack variable $\boldsymbol{\nu} \in \mathbb{R}^{T \times d}$ where $\boldsymbol{\nu}^t \in \mathbb{R}^d$ denotes the $t$-th row of matrix $\boldsymbol{\nu}$, and we can rewrite the original problem as a constrained optimization problem:

$$\hat{\boldsymbol{\phi}}, \hat{\boldsymbol{\mu}} = \operatorname*{arg\,min}_{\boldsymbol{\phi}, \boldsymbol{\mu}} -l(\boldsymbol{\phi}, \boldsymbol{\mu}) + \lambda \sum_{t=1}^{T-1} \|\boldsymbol{\nu}^{t+1} - \boldsymbol{\nu}^t\|_2$$
$$\text{subject to}\ \ \boldsymbol{\mu} = \boldsymbol{\nu}. \tag{2}$$

Then the augmented Lagrangian can be defined as

$$\mathcal{L}(\boldsymbol{\phi}, \boldsymbol{\mu}, \boldsymbol{\nu}, \boldsymbol{\rho}) = -l(\boldsymbol{\phi}, \boldsymbol{\mu}) + \lambda \sum_{t=1}^{T-1} \|\boldsymbol{\nu}^{t+1} - \boldsymbol{\nu}^t\|_2 + \operatorname{tr}[\boldsymbol{\rho}^\top (\boldsymbol{\mu} - \boldsymbol{\nu})] + \frac{\kappa}{2} \|\boldsymbol{\mu} - \boldsymbol{\nu}\|_F^2$$

where $\boldsymbol{\rho} \in \mathbb{R}^{T \times d}$ is the Lagrange multipliers and $\kappa > 0$ is the penalty parameter for the augmentation term. Let $\boldsymbol{w} = \kappa^{-1} \boldsymbol{\rho} \in \mathbb{R}^{T \times d}$ be the scaled dual variable, the augmented Lagrangian can be updated to

$$\mathcal{L}(\boldsymbol{\phi}, \boldsymbol{\mu}, \boldsymbol{\nu}, \boldsymbol{w}) = -l(\boldsymbol{\phi}, \boldsymbol{\mu}) + \lambda \sum_{t=1}^{T-1} \|\boldsymbol{\nu}^{t+1} - \boldsymbol{\nu}^t\|_2 + \frac{\kappa}{2} \|\boldsymbol{\mu} - \boldsymbol{\nu} + \boldsymbol{w}\|_F^2 - \frac{\kappa}{2} \|\boldsymbol{w}\|_F^2. \tag{3}$$

In practice, gradient descent may not work well for an objective function with Group Fused Lasso penalty. We further introduce two variables $(\boldsymbol{\gamma}, \boldsymbol{\beta}) \in \mathbb{R}^{1 \times d} \times \mathbb{R}^{(T-1) \times d}$ to ease the optimization, by converting it into a Group Lasso problem (Yuan & Lin, 2006). They are defined as

$$\boldsymbol{\gamma} = \boldsymbol{\nu}^1 \ \text{ and } \ \boldsymbol{\beta}_{t,\cdot} = \boldsymbol{\nu}^{t+1} - \boldsymbol{\nu}^t \ \ \forall\, t = 1, \ldots, T-1.$$

Reversely, the slack variable $\boldsymbol{\nu} \in \mathbb{R}^{T \times d}$ can be reconstructed as $\boldsymbol{\nu} = \mathbf{1}_{T,1}\boldsymbol{\gamma} + \boldsymbol{X}\boldsymbol{\beta}$, where $\boldsymbol{X}$ is a $T \times (T-1)$ design matrix with $\boldsymbol{X}_{ij} = 1$ for $i > j$ and 0 otherwise. Substituting the $\boldsymbol{\nu}$ in (3) with $(\boldsymbol{\gamma}, \boldsymbol{\beta})$, we have

$$\mathcal{L}(\boldsymbol{\phi}, \boldsymbol{\mu}, \boldsymbol{\gamma}, \boldsymbol{\beta}, \boldsymbol{w}) = -l(\boldsymbol{\phi}, \boldsymbol{\mu}) + \lambda \sum_{t=1}^{T-1} \|\boldsymbol{\beta}_{t,\cdot}\|_2 + \frac{\kappa}{2}\|\boldsymbol{\mu} - \mathbf{1}_{T,1}\boldsymbol{\gamma} - \boldsymbol{X}\boldsymbol{\beta} + \boldsymbol{w}\|_F^2 - \frac{\kappa}{2}\|\boldsymbol{w}\|_F^2.$$

Thus, we can derive the following Alternating Direction Method of Multipliers (ADMM) procedure (Boyd et al., 2011; Wang et al., 2019) to solve the constrained optimization problem in (2):

$$\boldsymbol{\phi}_{(a+1)}, \boldsymbol{\mu}_{(a+1)} = \underset{\boldsymbol{\phi},\boldsymbol{\mu}}{\arg\min} \ -l(\boldsymbol{\phi}, \boldsymbol{\mu}) + \frac{\kappa}{2}\|\boldsymbol{\mu} - \boldsymbol{\nu}_{(a)} + \boldsymbol{w}_{(a)}\|_F^2, \tag{4}$$

$$\boldsymbol{\gamma}_{(a+1)}, \boldsymbol{\beta}_{(a+1)} = \underset{\boldsymbol{\gamma},\boldsymbol{\beta}}{\arg\min} \ \lambda \sum_{t=1}^{T-1} \|\boldsymbol{\beta}_{t,\cdot}\|_2 + \frac{\kappa}{2}\|\boldsymbol{\mu}_{(a+1)} - \mathbf{1}_{T,1}\boldsymbol{\gamma} - \boldsymbol{X}\boldsymbol{\beta} + \boldsymbol{w}_{(a)}\|_F^2, \tag{5}$$

$$\boldsymbol{w}_{(a+1)} = \boldsymbol{\mu}_{(a+1)} - \boldsymbol{\nu}_{(a+1)} + \boldsymbol{w}_{(a)}, \tag{6}$$

where subscript $a$ denotes the current ADMM iteration. We recursively implement the three updates until a convergence criterion is satisfied. Throughout the paper, details about the implementation are provided in Appendix 7.4.

## 3.2 Parameters Update

### 3.2.1 Updating $\boldsymbol{\mu}$ and $\boldsymbol{\phi}$

In this section, we derive the updates for the prior and graph decoder parameters. Denote the objective function in (4) as $\mathcal{L}(\boldsymbol{\phi}, \boldsymbol{\mu})$. Setting the gradients of $\mathcal{L}(\boldsymbol{\phi}, \boldsymbol{\mu})$ with respect to the prior parameter $\boldsymbol{\mu}^t \in \mathbb{R}^d$ to zeros, we have the following:

**Proposition 1.** *The solution for $\boldsymbol{\mu}^t$ at an iteration of our proposed ADMM algorithm is a weighted sum:*

$$\boldsymbol{\mu}^t = \frac{1}{1+\kappa}\mathbb{E}_{P(\boldsymbol{z}^t|\boldsymbol{y}^t)}(\boldsymbol{z}^t) + \frac{\kappa}{1+\kappa}(\boldsymbol{\nu}^t - \boldsymbol{w}^t) \tag{7}$$

*between the conditional expectation of the latent variable under the posterior distribution $P(\boldsymbol{z}^t|\boldsymbol{y}^t)$ and the difference between the slack and the scaled dual variables. The term $\boldsymbol{w}^t \in \mathbb{R}^d$ denotes the $t$-th row of the scaled dual variable $\boldsymbol{w} \in \mathbb{R}^{T \times d}$. The derivation is provided in Appendix 7.1.*

Moreover, the gradient of $\mathcal{L}(\boldsymbol{\phi}, \boldsymbol{\mu})$ with respect to the graph decoder parameter $\boldsymbol{\phi}$ is calculated as

$$\nabla_{\boldsymbol{\phi}} \ \mathcal{L}(\boldsymbol{\phi}, \boldsymbol{\mu}) = -\sum_{t=1}^{T}\mathbb{E}_{P(\boldsymbol{z}^t|\boldsymbol{y}^t)}\Big(\nabla_{\boldsymbol{\phi}} \log P(\boldsymbol{y}^t|\boldsymbol{z}^t)\Big). \tag{8}$$

The parameter $\boldsymbol{\phi}$ can be updated efficiently through back-propagation.

Notably, calculating the solution in (7) and the gradient in (8) requires evaluating the conditional expectation under the posterior distribution $P(\boldsymbol{z}^t|\boldsymbol{y}^t) \propto P(\boldsymbol{y}^t|\boldsymbol{z}^t) \times P(\boldsymbol{z}^t)$. We employ Langevin Dynamics to sample from the posterior distribution, approximating the conditional expectations (Xie et al., 2017; 2018; Nijkamp et al., 2020; Pang et al., 2020). In particular, let subscript $\tau$ be the time step of the Langevin Dynamics and let $\delta$ be a small step size. Moving toward the gradient of the posterior with respect to the latent variable, the Langevin Dynamics to draw samples from the posterior distribution is achieved by iterating:

$$\boldsymbol{z}_{\tau+1}^t = \boldsymbol{z}_\tau^t + \delta\big[\nabla_{\boldsymbol{z}^t} \log P(\boldsymbol{z}^t|\boldsymbol{y}^t)\big] + \sqrt{2\delta}\boldsymbol{\epsilon}$$
$$= \boldsymbol{z}_\tau^t + \delta\big[\nabla_{\boldsymbol{z}^t} \log P_{\boldsymbol{\phi}}(\boldsymbol{y}^t|\boldsymbol{z}^t) - (\boldsymbol{z}_\tau^t - \boldsymbol{\mu}^t)\big] + \sqrt{2\delta}\boldsymbol{\epsilon} \tag{9}$$

where $\boldsymbol{\epsilon} \sim \mathcal{N}(\mathbf{0}, \boldsymbol{I}_d)$ is a random perturbation to the process. The derivation is provided in Appendix 7.2.

### 3.2.2  Updating $\gamma$ and $\beta$

In this section, we derive the update in (5), which is equivalent to solving a Group Lasso problem. In particular, we decompose the slack variable $\nu$ to work with $\gamma$ and $\beta$. With ADMM, the updates on $\gamma$ and $\beta$ do not require the observed network data $\{\boldsymbol{y}^t\}_{t=1}^T$.

By adapting the derivation in Bleakley & Vert (2011), we have the following for our proposed ADMM:

**Proposition 2. *[Bleakley & Vert, 2011]*** *The Group Lasso problem to update $\beta \in \mathbb{R}^{(T-1)\times d}$ is solved in a block coordinate descent manner, by iteratively applying the following equation to each row $t$:*

$$\beta_{t,\cdot} \leftarrow \frac{1}{\kappa \boldsymbol{X}_{\cdot,t}^\top \boldsymbol{X}_{\cdot,t}} \left(1 - \frac{\lambda}{\|\boldsymbol{b}_t\|_2}\right)_+ \boldsymbol{b}_t \tag{10}$$

*where $(\cdot)_+ = \max(\cdot, 0)$ and*

$$\boldsymbol{b}_t = \kappa \boldsymbol{X}_{\cdot,t}^\top (\boldsymbol{\mu}_{(a+1)} + \boldsymbol{w}_{(a)} - \mathbf{1}_{T,1}\gamma - \boldsymbol{X}_{\cdot,-t}\beta_{-t,\cdot}).$$

*The derivation is provided in Appendix 7.3.*

The convergence of the procedure can be monitored by the Karush-Kuhn-Tucker conditions: for all $\beta_{t,\cdot} \neq \mathbf{0}$,

$$\lambda \frac{\beta_{t,\cdot}}{\|\beta_{t,\cdot}\|_2} - \kappa \boldsymbol{X}_{\cdot,t}^\top (\boldsymbol{\mu}_{(a+1)} + \boldsymbol{w}_{(a)} - \mathbf{1}_{T,1}\gamma - \boldsymbol{X}\beta) = \mathbf{0},$$

and for all $\beta_{t,\cdot} = \mathbf{0}$,

$$\|-\kappa \boldsymbol{X}_{\cdot,t}^\top (\boldsymbol{\mu}_{(a+1)} + \boldsymbol{w}_{(a)} - \mathbf{1}_{T,1}\gamma - \boldsymbol{X}\beta)\|_2 \leq \lambda.$$

Lastly, for any $\beta$, the minimum in $\gamma \in \mathbb{R}^{1\times d}$ is achieved at

$$\gamma = (1/T)\mathbf{1}_{1,T} \cdot (\boldsymbol{\mu}_{(a+1)} + \boldsymbol{w}_{(a)} - \boldsymbol{X}\beta).$$

In summary, the procedure to solve the problem in (2) via ADMM is presented in Algorithm 1. The steps to transform between $\nu$ and $(\gamma, \beta)$ within an ADMM iteration are omitted for succinctness. The complexity of the proposed algorithm is at least of order $O\big(A(Tsl + BT + D(T-1))\big)$ with additional gradient calculation for neural networks in the sub-routines. Specifically, for each of the $A$ iterations of ADMM, we update the prior parameters $\boldsymbol{\mu}^t$ for all $T$ time points, and each update involves $l$ steps of MCMC for $s$ samples. Then we calculate the gradients for neural networks over the $T$ time points and run $B$ iterations of Adam optimizer. Lastly, we run $D$ iterations of block coordinate descent for the $T-1$ sequential differences.

## 4  Change Point Localization and Model Selection

### 4.1  Change Point Localization

In this section, we provide two effective methods to localize the change points after parameter learning, and they can be used for different purposes. For the first approach, we can resort to the prior distribution where $\boldsymbol{z}^t \sim \mathcal{N}(\boldsymbol{\mu}^t, \boldsymbol{I}_d)$. When no change occurs or $\boldsymbol{\mu}^t - \boldsymbol{\mu}^{t-1} = \mathbf{0}$, we have $\boldsymbol{z}^t - \boldsymbol{z}^{t-1} \sim \mathcal{N}(\mathbf{0}, 2\boldsymbol{I}_d)$ and

$$u^t := \frac{1}{2}(\boldsymbol{z}^t - \boldsymbol{z}^{t-1})^\top (\boldsymbol{z}^t - \boldsymbol{z}^{t-1}) \sim \chi_d^2.$$

Furthermore, the mean of $u^t$ over $m$ samples follows a Gamma distribution:

$$\bar{u}^t \sim \Gamma(\theta = \frac{2}{m}, \xi = \frac{md}{2})$$

where $\theta$ and $\xi$ are the respective scale and shape parameters.

---

**Algorithm 1** Latent Space Group Fused Lasso

---

1: **Input**: learning iterations $A, B, D$, tuning parameter $\lambda$, penalty parameter $\kappa$, learning rates $\eta$, observed data $\{\boldsymbol{y}^t\}_{t=1}^T$, initialization $\{\boldsymbol{\phi}_{(1)}, \boldsymbol{\mu}_{(1)}, \boldsymbol{\gamma}_{(1)}, \boldsymbol{\beta}_{(1)}, \boldsymbol{w}_{(1)}\}$
2: **for** $a = 1, \cdots, A$ **do**
3:     **for** $t = 1, \cdots, T$ **do**
4:         draw $s$ samples $\boldsymbol{z}_1^t, \ldots, \boldsymbol{z}_s^t$ from $P(\boldsymbol{z}^t | \boldsymbol{y}^t)$ according to (9)
5:         $\boldsymbol{\mu}_{(a+1)}^t = \frac{1}{1+\kappa}(s^{-1}\sum_{i=1}^s \boldsymbol{z}_i^t) + \frac{\kappa}{1+\kappa}(\boldsymbol{\nu}^t - \boldsymbol{w}^t)$
6:     **end for**
7:     **for** $b = 1, \ldots, B$ **do**
8:         $\boldsymbol{\phi}_{(b+1)} = \boldsymbol{\phi}_{(b)} - \eta \times \nabla_{\boldsymbol{\phi}} \, \mathcal{L}(\boldsymbol{\phi}, \boldsymbol{\mu})$
9:     **end for**
10:     Set $\tilde{\boldsymbol{\gamma}}^{(1)} = \boldsymbol{\gamma}_{(a)}$ and $\tilde{\boldsymbol{\beta}}^{(1)} = \boldsymbol{\beta}_{(a)}$
11:     **for** $d = 1, \ldots, D$ **do**
12:         **for** $t = 1, \ldots, T-1$ **do**
13:             Let $\tilde{\boldsymbol{\beta}}_{t,\cdot}^{(d+1)}$ be updated according to (10)
14:         **end for**
15:         $\tilde{\boldsymbol{\gamma}}^{(d+1)} = (1/T)\mathbf{1}_{1,T} \cdot (\boldsymbol{\mu}_{(a+1)} + \boldsymbol{w}_{(a)} - \boldsymbol{X}\tilde{\boldsymbol{\beta}}^{(d+1)})$
16:     **end for**
17:     Set $\boldsymbol{\gamma}_{(a+1)} = \tilde{\boldsymbol{\gamma}}^{(d+1)}$ and $\boldsymbol{\beta}_{(a+1)} = \tilde{\boldsymbol{\beta}}^{(d+1)}$
18:     $\boldsymbol{w}_{(a+1)} = \boldsymbol{\mu}_{(a+1)} - \boldsymbol{\nu}_{(a+1)} + \boldsymbol{w}_{(a)}$
19: **end for**
20: $\hat{\boldsymbol{\mu}} \leftarrow \boldsymbol{\mu}_{(a+1)}$
21: **Output**: learned prior parameters $\hat{\boldsymbol{\mu}}$

---

As we capture the structural changes in the latent space, we can draw samples from the learned priors to reflect the sequential changes. In particular, for a time point $t$, we sample $\hat{\boldsymbol{z}}^t - \hat{\boldsymbol{z}}^{t-1}$ from $\mathcal{N}(\hat{\boldsymbol{\mu}}^t - \hat{\boldsymbol{\mu}}^{t-1}, 2\boldsymbol{I}_d)$, and we perform the same transformation:

$$v^t := \frac{1}{2}(\hat{\boldsymbol{z}}^t - \hat{\boldsymbol{z}}^{t-1})^\top(\hat{\boldsymbol{z}}^t - \hat{\boldsymbol{z}}^{t-1}).$$

Then we compare the mean of $v^t$ over $m$ samples with a quantile:

$$P(\bar{v}^t > q_{\text{thr}}) = 1 - \frac{\alpha}{T-1} \tag{11}$$

where $q_{\text{thr}}$ is the $1 - \alpha/(T-1)$ quantile of the Gamma distribution for $\bar{u}^t$ when no change occurs. We consider the time point $t$ with $\bar{v}^t > q_{\text{thr}}$ as the detected change point.

For the second approach, we can utilize the localizing method from Kei et al. (2023), which is more robust in practice, as compared in the simulation study of Section 5.1. First, we calculate the differences between consecutive time points in $\hat{\boldsymbol{\mu}} \in \mathbb{R}^{T \times d}$ as

$$\Delta\hat{\boldsymbol{\mu}}^t = \|\boldsymbol{\mu}^t - \boldsymbol{\mu}^{t-1}\|_2 \quad \forall \, t \in [2, T].$$

Then we standardize the differences as

$$\Delta\hat{\boldsymbol{\zeta}}^t = \frac{\Delta\hat{\boldsymbol{\mu}}^t - \text{median}(\Delta\hat{\boldsymbol{\mu}})}{\text{std}(\Delta\hat{\boldsymbol{\mu}})} \quad \forall \, t \in [2, T] \tag{12}$$

and construct a data-driven threshold defined as

$$\mathcal{T}_{\text{thr}} := \text{mean}(\Delta\hat{\boldsymbol{\zeta}}) + \mathcal{Z}_q \times \text{std}(\Delta\hat{\boldsymbol{\zeta}}) \tag{13}$$

where $\mathcal{Z}_q$ is the $q\%$ quantile of standard Normal distribution. Finally, we declare a change point $C_k$ when $\Delta\hat{\boldsymbol{\zeta}}^{C_k} > \mathcal{T}_{\text{thr}}$.

The data-driven threshold in (13) is intuitive, as the standardized differences $\Delta\hat{\boldsymbol{\zeta}}$ between two consecutive change points are close to zeros, while the differences that are at the change points are substantially greater than zeros. When traced in a plot over time $t$, the $\Delta\hat{\boldsymbol{\zeta}}$ can exhibit the magnitude of structural changes, and the threshold that deviates from the mean provides a reasonable cut-off value for the standardized differences, as demonstrated in Figures 5 and 6. In summary, the localizing method derived from the prior distribution has a statistical justification, while the localizing method with the data-driven threshold is more robust for different types of network data in practice.

## 4.2 Model Selection

The optimization problem in (2) involves a tuning parameter that can yield different sets of detected change points when it is varied. In this work, we use Cross-Validation to select $\lambda$. In particular, we split the original time series of graphs into training and testing sets: the training set consists of graphs at odd indexed time points and the testing set consists of graphs at even indexed time points. Fixed on a specific $\lambda$ value, we learn the model parameters with the training set, and we evaluate the learned model with the testing set.

For a list of $\lambda$ values, we choose the $\lambda$ giving the maximal log-likelihood on the testing set. Note that the log-likelihood is approximated by Monte Carlo samples $\{\boldsymbol{z}_u^t\}_{u=1}^s$ drawn from the prior distribution $P(\boldsymbol{z}^t)$ as

$$\sum_{t=1}^T \log P(\boldsymbol{y}^t) \approx \sum_{t=1}^T \log\Big[\frac{1}{s}\sum_{u=1}^s \big[\prod_{(i,j)\in\mathbb{Y}} P_\phi(\boldsymbol{y}_{ij}^t|\boldsymbol{z}_u^t)\big]\Big].$$

Further computational details are discussed in Appendix 7.4. Anchored on the selected $\lambda$ value, we learn the model parameters again with the full data, resulting the final set of detected change points.

## 5 Simulated and Real Data Experiments

In this section, we implement the proposed method on simulated and real data. To evaluate the performance for simulated data, we use three standard metrics in the literature that focus on the number of change points, the time gap between the true and detected change points, and the coverage between the segmented time intervals. The first metric is the absolute error $|\hat{K} - K|$, where $\hat{K}$ and $K$ are the respective numbers of the detected and true change points. The second metric described in Madrid Padilla et al. (2021) is the one-sided Hausdorff distance, which is defined as

$$d(\hat{\mathcal{C}}|\mathcal{C}) = \max_{c\in\mathcal{C}} \min_{\hat{c}\in\hat{\mathcal{C}}} |\hat{c} - c|$$

where $\hat{\mathcal{C}}$ and $\mathcal{C}$ are the respective sets of detected and true change points. Also, we report the reversed one-sided Hausdorff distance $d(\mathcal{C}|\hat{\mathcal{C}})$. By convention, when $\hat{\mathcal{C}} = \emptyset$, we let $d(\hat{\mathcal{C}}|\mathcal{C}) = \infty$ and $d(\mathcal{C}|\hat{\mathcal{C}}) = -\infty$. The last metric described in van den Burg & Williams (2020) is the coverage of a partition $\mathcal{G}$ by another partition $\mathcal{G}'$, which is defined as

$$C(\mathcal{G}, \mathcal{G}') = \frac{1}{T}\sum_{\mathcal{A}\in\mathcal{G}} |\mathcal{A}| \cdot \max_{\mathcal{A}'\in\mathcal{G}'} \frac{|\mathcal{A}\cap\mathcal{A}'|}{|\mathcal{A}\cup\mathcal{A}'|}$$

with $\mathcal{A}, \mathcal{A}' \subseteq [1, T]$. The $\mathcal{G}$ and $\mathcal{G}'$ are collections of intervals between consecutive change points for the respective true and detected change points.

## 5.1 Simulation Study

We simulate dynamic graphs from three scenarios to compare the performance of the proposed and competing methods: Separable Temporal Exponential Random Graph Model, Stochastic Block Model, and Recurrent Neural Network. For each scenario with different numbers of nodes $n \in \{50, 100\}$, we simulate 10 Monte Carlo trials of directed dynamic graphs with time span $T = 100$. The true change points are located at $t = 26, 51, 76$, so the number of change points $K = 3$. Moreover, the $K + 1 = 4$ intervals in the partition $\mathcal{G}$

are $\mathcal{A}_1 = [1, \ldots, 25]$, $\mathcal{A}_2 = [26, \ldots, 50]$, $\mathcal{A}_3 = [51, \ldots, 75]$, and $\mathcal{A}_4 = [76, \ldots, 100]$. In each specification, we report the means and standard deviations over 10 Monte Carlo trials for the evaluation metrics. $\text{CPDlatent}_N$ denotes our proposed approach with the data-driven threshold in (13), using 90% quantile from standard Normal distribution. We let the latent dimensions $d = 10$ and $k = 5$ for the graph decoder. $\text{CPDlatent}_G$ denotes our proposed approach with the localizing method in (11), using $\alpha = 0.01$ from Gamma distribution. We let the latent dimensions $k = 10$ and $d = n/10$ for the graph decoder. The number of samples drawn from the Gamma distribution is $m = 500$ when $d = 10$ and $m = 1000$ when $d = 5$.

Three competitors, gSeg (Chen & Zhang, 2015), kerSeg (Song & Chen, 2022b), and CPDstergm (Kei et al., 2023), are provided for comparison. The gSeg utilizes graph-based scan statistics and kerSeg employs a kernel-based framework to test the partition before and after a change point. The CPDstergm fits a STERGM with user-specified network statistics to detect change points based on the model parameters. For CPDstergm, we first use two network statistics, edge count and mutuality, in both formation and dissolution models to let $p = 4$. We then add one more network statistic, number of triangles, to let $p = 6$ as another specification. For gSeg, we use the minimum spanning tree to construct the similarity graph, with the approximated p-value of the original edge-count scan statistic, and we set $\alpha = 0.05$. For kerSeg, we use the approximated p-value of the $\text{fGKCP}_1$, and we set $\alpha = 0.001$. Moreover, we use networks (nets.) and network statistics (stats.) as two types of input data to gSeg and kerSeg. Throughout, we choose these settings because they produce good performance on average for the competitors. Changing the settings can enhance their performance on some specifications, while severely jeopardizing their performance on other specifications.

**Scenario 1: Separable Temporal Exponential Random Graph Model**

In this scenario, we apply time-homogeneous Separable Temporal Exponential Random Graph Model (STERGM) between change points to generate sequences of dynamic networks (Krivitsky & Handcock, 2014). We use three network statistics, edge count, mutuality, and number of triangles, in both formation (F) and dissolution (D) models. The $p = 6$ parameters for each time point $t$ are

$$\boldsymbol{\theta}_F^t, \boldsymbol{\theta}_D^t = \begin{cases} -2, \ 2, \ -2, \ -1, \ 2, \ 1, & t \in \mathcal{A}_1 \cup \mathcal{A}_3 \setminus 1, \\ -1.5, \ 1, \ -1, \ 2, \ 1, \ 1.5, & t \in \mathcal{A}_2 \cup \mathcal{A}_4. \end{cases}$$

Figure 2 exhibits examples of generated networks. Visually, STERGM produces adjacency matrices that are sparse, which is often the case in real world social networks.

Table 1 displays the means and standard deviations of the evaluation metrics for comparison. Since the networks are directly sampled from STERGM, the CPDstergm method with correctly specified network statistics ($p = 6$) achieves the best result, in terms of greater converge of the intervals. However, when the network statistics are mis-specified ($p = 4$), the performance of CPDstergm is worsened, with greater gaps between the true and detected change points. Also, using either networks (nets.) or network statistics (stats.) cannot improve the performance of gSeg and kerSeg methods: the binary search approach tend to detect excessive number of change points. Our CPDlatent method, without the need of specifying network statistics, can achieve relatively good performance on average.

**Scenario 2: Stochastic Block Model**

In this scenario, we use Stochastic Block Model (SBM) to generate sequences of dynamic networks, and we impose a time-dependent mechanism in the generation process as in Madrid Padilla et al. (2022). Two probability matrices $\boldsymbol{P}, \boldsymbol{Q} \in [0,1]^{n \times n}$ are constructed and they are defined as

$$\boldsymbol{P}_{ij} = \begin{cases} 0.5, & i, j \in \mathcal{B}_l, \ l \in [3], \\ 0.3, & \text{otherwise,} \end{cases} \quad \text{and} \quad \boldsymbol{Q}_{ij} = \begin{cases} 0.45, & i, j \in \mathcal{B}_l, \ l \in [3], \\ 0.2, & \text{otherwise,} \end{cases}$$

where $\mathcal{B}_1, \mathcal{B}_2, \mathcal{B}_3$ are evenly sized clusters that form a partition of $\{1, \ldots, n\}$. Then a sequence of matrices $\boldsymbol{E}^t \in [0,1]^{n \times n}$ are arranged for $t = 1, \ldots, T$ such that

$$\boldsymbol{E}_{ij}^t = \begin{cases} \boldsymbol{P}_{ij}, & t \in \mathcal{A}_1 \cup \mathcal{A}_3, \\ \boldsymbol{Q}_{ij}, & t \in \mathcal{A}_2 \cup \mathcal{A}_4. \end{cases}$$

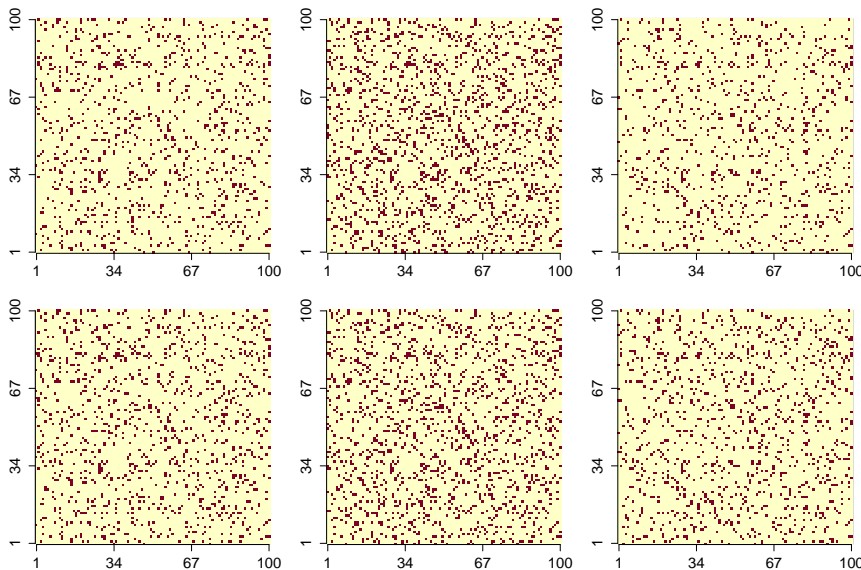

Figure 2: Examples of networks generated from STERGM with $n = 100$. In the first row, from left to right, each plot corresponds to the network at $t = 25, 50, 75$ respectively. In the second row, from left to right, each plot corresponds to the network at $t = 26, 51, 76$ respectively (the change points).

Table 1: Means (standard deviations) of evaluation metrics for dynamic graphs simulated from STERGM. The best coverage metric is bolded.

| $n$ | Method | $|\hat{K} - K| \downarrow$ | $d(\hat{\mathcal{C}}|\mathcal{C}) \downarrow$ | $d(\mathcal{C}|\hat{\mathcal{C}}) \downarrow$ | $C(\mathcal{G}, \mathcal{G}') \uparrow$ |
|---|---|---|---|---|---|
| 50 | CPDlatent$_N$ | 0.1 (0.3) | 4.3 (5.7) | 2.6 (1.3) | 90.87% |
| | CPDlatent$_G$ | 0.4 (0.6) | 4.2 (6.9) | 3.4 (3.4) | 90.97% |
| | CPDstergm$_{p=4}$ | 1.5 (0.8) | 11.7 (7.5) | 10.5 (2.3) | 67.68% |
| | CPDstergm$_{p=6}$ | 0.2 (0.4) | 1.6 (1.2) | 3 (3.5) | **91.54**% |
| | gSeg (nets.) | 12.3 (0.5) | 0 (0) | 19 (0) | 27.90% |
| | gSeg (stats.) | 15.8 (0.7) | 1.5 (0.5) | 20.1 (0.3) | 24.55% |
| | kerSeg (nets.) | 9.7 (0.9) | 1.4 (0.9) | 17.9 (1.2) | 37.62% |
| | kerSeg (stats.) | 9.4 (0.7) | 3.9 (1.3) | 18 (1.8) | 35.86% |
| 100 | CPDlatent$_N$ | 0 (0) | 3.9 (1.3) | 3.9 (1.3) | 91.33% |
| | CPDlatent$_G$ | 0.7 (1.3) | 3.1 (1.3) | 6.0 (4.0) | 88.55% |
| | CPDstergm$_{p=4}$ | 0.7 (0.6) | 21.9 (10.3) | 7.6 (4.3) | 67.21% |
| | CPDstergm$_{p=6}$ | 0 (0) | 1.1 (0.3) | 1.1 (0.3) | **94.01**% |
| | gSeg (nets.) | 12 (0) | 0 (0) | 19 (0) | 28.00% |
| | gSeg (stats.) | 14.5 (2.3) | 3.3 (3.6) | 20.2 (0.4) | 26.13% |
| | kerSeg (nets.) | 9.3 (0.8) | 1 (0) | 17.7 (0.6) | 37.62% |
| | kerSeg (stats.) | 8.5 (0.8) | 4.5 (1.4) | 17.3 (1.7) | 36.92% |

Lastly, the networks are generated with $\rho = 0.5$ as a time-dependent mechanism. For $t = 1, \ldots, T - 1$, we let $\boldsymbol{y}_{ij}^1 \sim \text{Bernoulli}(\boldsymbol{E}_{ij}^1)$ and

$$\boldsymbol{y}_{ij}^{t+1} \sim \begin{cases} \text{Bernoulli}\big(\rho(1 - \boldsymbol{E}_{ij}^{t+1}) + \boldsymbol{E}_{ij}^{t+1}\big), & \boldsymbol{y}_{ij}^t = 1, \\ \text{Bernoulli}\big((1 - \rho)\boldsymbol{E}_{ij}^{t+1}\big), & \boldsymbol{y}_{ij}^t = 0. \end{cases}$$

²¹² With $\rho > 0$, the probability to form an edge for $i, j$ becomes greater at time $t + 1$ when there exists an edge
²¹³ at time $t$, and the probability becomes smaller when there does not exist an edge at time $t$. Figure 3 exhibits
²¹⁴ examples of generated networks. Visually, SBM produces adjacency matrices with block structures, where
²¹⁵ mutuality serves as an important pattern for the homophily within groups.

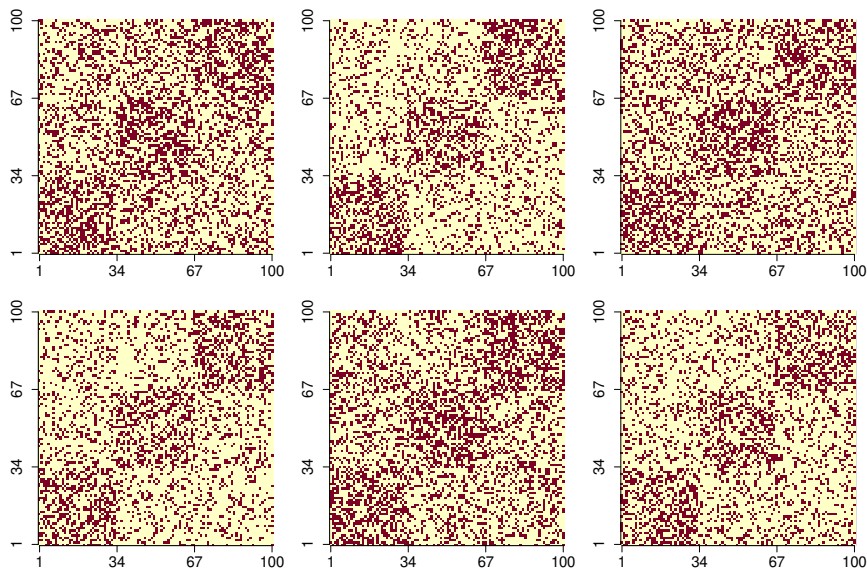

Figure 3: Examples of networks generated from SBM with $n = 100$. In the first row, from left to right, each
plot corresponds to the network at $t = 25, 50, 75$ respectively. In the second row, from left to right, each plot
corresponds to the network at $t = 26, 51, 76$ respectively (the change points).

²¹⁶ Table 2 displays the means and standard deviations of the evaluation metrics for comparison. As expected,
²¹⁷ both CPDstergm methods with $p = 4$ and $p = 6$ that utilize the mutuality as a network statistic for the
²¹⁸ detection can achieve good results, in terms of greater converge of the intervals. Furthermore, using network
²¹⁹ statistics (stats.) for both gSeg and kerSeg methods can improve their performance, comparing to using
²²⁰ networks (nets.) as input data. Lastly, our CPDlatent method, which infers the features in latent space that
²²¹ induce the structural changes, achieves the best result for networks with block structures.

²²² **Scenario 3: Recurrent Neural Networks**

In this scenario, we use Recurrent Neural Networks (RNN) to generate sequences of dynamic networks.
Specifically, we sample latent variables from pre-defined priors, and we initialize the RNN with uniform
weights. The graphs are then generated by the matrix multiplication defined in Section 2.2, using the output
of RNN. The parameters for the pre-defined priors are

$$z^t \sim \begin{cases} \mathcal{N}(-\mathbf{1}, \ 0.1\boldsymbol{I}_d), & t \in \mathcal{A}_1 \cup \mathcal{A}_3, \\ \mathcal{N}(\mathbf{5}, \ 0.1\boldsymbol{I}_d), & t \in \mathcal{A}_2 \cup \mathcal{A}_4. \end{cases}$$

²²³ Similar to the previous two scenarios, the simulation using RNN also imposes a time-dependent mechanism
²²⁴ across dynamic networks. Figure 4 exhibits examples of generated networks. Visually, RNN produces
²²⁵ adjacency matrices that are dense, and no discernible pattern can be noticed.

²²⁶ Table 3 displays the means and standard deviations of the evaluation metrics for comparison. Because no
²²⁷ structural pattern or suitable network statistics can be determined a priori, neither CPDstergm method with
²²⁸ $p = 4$ nor with $p = 6$ can detect the change points accurately. Likewise, both gSeg and kerSeg methods that
²²⁹ utilize the mis-specified network statistics (stats.) cannot produce satisfactory performance. Notably, the
²³⁰ kerSeg method that exploits the features in high dimension with networks (nets.) instead of user-specified

Table 2: Means (stds.) of evaluation metrics for dynamic networks simulated from SBM. The best coverage metric is bolded.

| $n$ | Method | $|\hat{K} - K|\downarrow$ | $d(\hat{\mathcal{C}}|\mathcal{C})\downarrow$ | $d(\mathcal{C}|\hat{\mathcal{C}})\downarrow$ | $C(\mathcal{G},\mathcal{G}')\uparrow$ |
|---|---|---|---|---|---|
| 50 | CPDlatent$_N$ | 0 (0) | 0.1 (0.3) | 0.1 (0.3) | **99.80**% |
| | CPDlatent$_G$ | 0.3 (0.6) | 0.1 (0.3) | 3.1 (6.2) | 96.70% |
| | CPDstergm$_{p=4}$ | 0.1 (0.3) | 1 (0) | 2.4 (4.2) | 97.04% |
| | CPDstergm$_{p=6}$ | 0.3 (0.5) | 1 (0) | 4.6 (5.6) | 94.74% |
| | gSeg (nets.) | 12.9 (1.8) | 0 (0) | 19.4 (0.8) | 27.20% |
| | gSeg (stats.) | 2.2 (0.7) | Inf (na) | $-$Inf (na) | 49.21% |
| | kerSeg (nets.) | 6.4 (1.4) | 0 (0) | 16.6 (2.0) | 45.50% |
| | kerSeg (stats.) | 0.9 (1.2) | 0 (0) | 5.6 (6.8) | 93.50% |
| 100 | CPDlatent$_N$ | 0.1 (0.3) | 0.1 (0.3) | 1.3 (3.6) | **98.60**% |
| | CPDlatent$_G$ | 0.5 (0.7) | 0.2 (0.4) | 5.1 (6.1) | 94.81% |
| | CPDstergm$_{p=4}$ | 0 (0) | 1 (0) | 1 (0) | 98.04% |
| | CPDstergm$_{p=6}$ | 0 (0) | 1 (0) | 1 (0) | 98.04% |
| | gSeg (nets.) | 12.3 (0.9) | 0 (0) | 19 (0) | 27.80% |
| | gSeg (stats.) | 2 (0.4) | Inf (na) | $-$Inf (na) | 55.75% |
| | kerSeg (nets.) | 6 (0.8) | 0 (0) | 15.2 (2.0) | 47.00% |
| | kerSeg (stats.) | 0.9 (0.7) | 0 (0) | 9.6 (7.6) | 93.40% |

Figure 4: Examples of networks generated from RNN with $n = 100$. In the first row, from left to right, each plot corresponds to the network at $t = 25, 50, 75$ respectively. In the second row, from left to right, each plot corresponds to the network at $t = 26, 51, 76$ respectively (the change points).

network statistics (stats.) can deliver a good result. Lastly, our CPDlatent method that first infers the graph level representations from the complex network structures and then utilize them to detect the change points yields the best result.

Table 3: Means (stds.) of evaluation metrics for dynamic networks simulated from RNN. The best coverage metric is bolded.

| $n$ | Method | $|\hat{K} - K| \downarrow$ | $d(\hat{\mathcal{C}}|\mathcal{C}) \downarrow$ | $d(\mathcal{C}|\hat{\mathcal{C}}) \downarrow$ | $C(\mathcal{G}, \mathcal{G}') \uparrow$ |
|---|---|---|---|---|---|
| 50 | CPDlatent$_N$ | 0 (0) | 1.8 (0.7) | 1.8 (0.7) | **94.77**% |
| | CPDlatent$_G$ | 0.3 (0.6) | 1.7 (0.6) | 3.2 (3.0) | 93.04% |
| | CPDstergm$_{p=4}$ | 2.0 (1.7) | 6.0 (7.7) | 15.2 (4.9) | 72.10% |
| | CPDstergm$_{p=6}$ | 1.0 (0.4) | 18.5 (9.4) | 14.3 (2.9) | 60.25% |
| | gSeg (nets.) | 2.3 (0.6) | Inf (na) | $-$Inf (na) | 29.42% |
| | gSeg (stats.) | 2.9 (0.3) | Inf (na) | $-$Inf (na) | 2.47% |
| | kerSeg (nets.) | 1.5 (0.9) | 1.4 (0.7) | 5.3 (3.3) | 89.25% |
| | kerSeg (stats.) | 2.8 (0.4) | Inf (na) | $-$Inf (na) | 9.89% |
| 100 | CPDlatent$_N$ | 0 (0) | 2.5 (0.7) | 2.5 (0.7) | 91.96% |
| | CPDlatent$_G$ | 0.2 (0.6) | 2.1 (0.7) | 2.8 (1.8) | **92.34**% |
| | CPDstergm$_{p=4}$ | 2.0 (1.4) | 10.6 (8.0) | 14.1 (3.1) | 60.37% |
| | CPDstergm$_{p=6}$ | 1.2 (1.3) | 20.6 (12.6) | 15.2 (5.9) | 53.21% |
| | gSeg (nets.) | 3 (0) | Inf (na) | $-$Inf (na) | 0% |
| | gSeg (stats.) | 2.9 (0.3) | Inf (na) | $-$Inf (na) | 4.27% |
| | kerSeg (nets.) | 1.4 (0.7) | 1.9 (0.7) | 5.4 (1.9) | 88.95% |
| | kerSeg (stats.) | 3 (0) | Inf (na) | $-$Inf (na) | 0% |

## 5.2 MIT Cellphone Data

The Massachusetts Institute of Technology (MIT) cellphone data (Eagle & Pentland, 2006) depicts human interactions via phone call activities among $n = 96$ participants spanning $T = 232$ days. An edge $\boldsymbol{y}_{ij}^t = 1$ in the constructed networks indicates that participant $i$ and participant $j$ had made phone calls on day $t$, and $\boldsymbol{y}_{ij}^t = 0$ otherwise. The data ranges from 2004-09-15 to 2005-05-04, covering the winter break in the MIT academic calendar.

We detect the change points with our proposed method using the data-driven threshold from standard Normal distribution, and we use network statistics as input data to the competitor methods. Specifically, we use the number of edges, isolates, and triangles to capture the frequency of connections, the sparsity of social interaction, and the transitive association among friends, respectively. Figure 5 displays $\Delta\hat{\boldsymbol{\zeta}}$ of Equation (12), and the detected change points from our method and competitor methods. Furthermore, Table 4 provides a list of potential events, aligning with the detected change points from our method.

Without specifying the structural changes to search for, our method can punctually detect the beginning of the winter break, which is the major event that alters the interaction among participants. Similar to the competitors, we have detected a spike on 2004-10-23, corresponding to the annual sponsor meeting that occurred on 2004-10-21. More than two-thirds of the participants have attended the meeting, focusing on achieving project goals throughout the week (Eagle & Pentland, 2006). Moreover, we have detected other change points related to national holidays and spring break.

## 5.3 Enron Email Data

The Enron email data, analyzed by Priebe et al. (2005), Park et al. (2012), and Peel & Clauset (2015), portrays communication among employees before the collapse of a giant energy company. The dynamic network data consists of $T = 100$ weekly networks, ranging from 2000-06-05 to 2002-05-06 for $n = 100$ employees. We detect the change points with our proposed method using the data-driven threshold from standard Normal distribution, and we use the same network statistics described in Section 5.2 to the competitor methods.

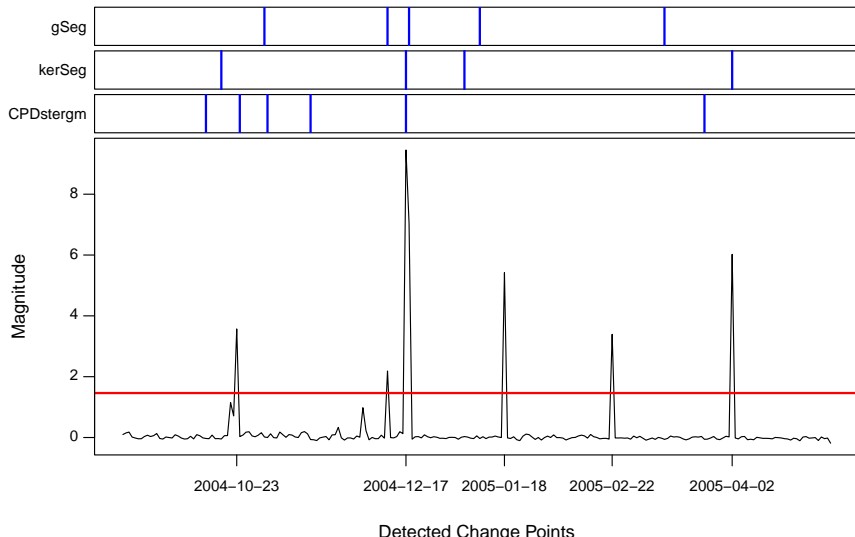

Figure 5: Detected change points from the proposed and competitor (blue) methods on the MIT Cellphone Data. The threshold (red horizontal line) is calculated by (13) with $\mathcal{Z}_{0.9}$.

Table 4: Potential nearby events aligned with the detected change points from our proposed method on the MIT cellphone data.

| Detected change points | Potential nearby events |
| --- | --- |
| 2004-10-23 | 2004-10-21 Sponsor meeting |
| 2004-12-17 | 2004-12-18 to 2005-01-02 Winter break |
| 2005-01-18 | 2005-01-17 Martin Luther King Day |
| 2005-02-22 | 2005-02-21 Presidents Day |
| 2005-04-02 | 2005-03-21 to 2005-03-25 Spring break |

Figure 6 displays $\Delta\hat{\zeta}$ of Equation (12), and the detected change points from our method and competitor methods. Furthermore, Table 5 provides a list of potential events, aligning with the detected change points from our method.

In 2001, Enron underwent a multitude of major and overlapping incidents, making it difficult to associate the detected change points with specific real world events. Yet, as our proposed method detects the results over the two-year time frame, four crucial change points are detected for interpretation. Throughout 2000, Enron orchestrated rolling blackouts, causing staggering surges in electricity prices that peaked at twenty times the standard rate. The situation worsened when the Federal Energy Regulatory Commission (FERC) exonerated Enron of wrongdoing by the end of 2000. During a public appearance in June 2001, the CEO is physically confronted by an activist in protest against Enron's role in the energy crisis. Amid the turmoil, an employee meeting took place in September 2001, where the CEO reassured employees that Enron's stock was a good buy and the company's accounting methods were legal and appropriate. Following the employee meeting, the stock saw a brief surge before continuing its sharp decline. Three months later, pressured by Wall Street analysts and the revelation of the scandals, Enron filed for bankruptcy and the largest energy company in the U.S. fell apart.

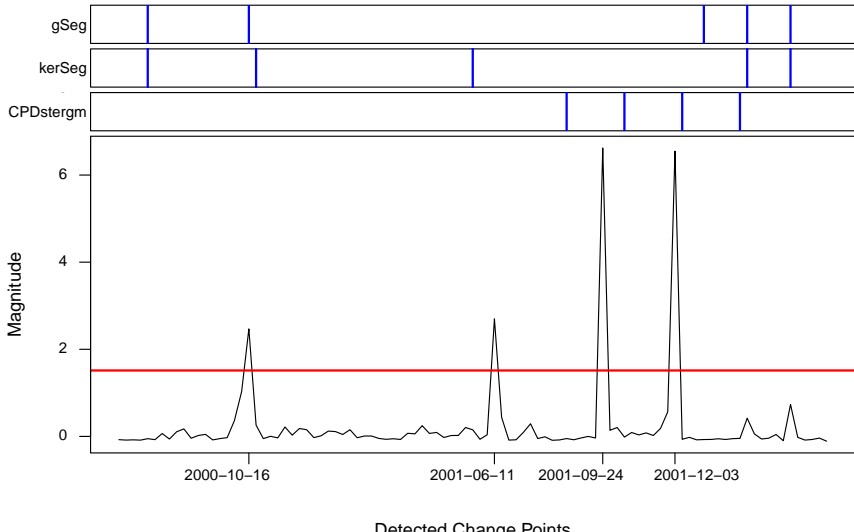

Figure 6: Detected change points from the proposed and competitor (blue) methods on the Enron email data. The threshold (red horizontal line) is calculated by (13) with $\mathcal{Z}_{0.9}$.

Table 5: Potential nearby events aligned with the detected change points from our proposed method on the Enron email data.

| Detected change points | Potential nearby events |
| --- | --- |
| 2000-10-16 | 2000-11-01 FERC exonerated Enron |
| 2001-06-11 | 2001-06-21 CEO publicly confronted |
| 2001-09-24 | 2001-09-26 Employee meeting |
| 2001-12-03 | 2001-12-02 Enron filed for bankruptcy |

## 6 Discussion

This paper proposes a generative model to detect change points in dynamic graphs. Intrinsically, dynamic networks are complex due to dyadic and temporal dependencies. Learning low dimensional graph representations can extract useful features to facilitate change point detection in dynamic graphs. We impose prior distributions to the graph representations, and the priors for the latent space are learned from the data as empirical Bayes. The optimization problem with Group Fused Lasso penalty is solved via ADMM, and generative model is demonstrated to be useful for change point detection.

Several extensions to our proposed framework are possible for future development. Besides binary networks, relations by nature have degree of strength, which are denoted by generic values. Also, nodal and dyadic attributes are important components in network data. Hence, models that can generate weighted edges, as well as nodal and dyadic attributes, can capture more information about the network dynamics (Fellows & Handcock, 2012; Krivitsky, 2012; Simonovsky & Komodakis, 2018). Furthermore, the number of nodes and their attributes are subjected to change over time. Extending the framework to allow the network size to change and to detect vertex level anomalies can provide granular insights in addition to graph level changes (Simonovsky & Komodakis, 2018; Shen et al., 2023). Similarly, improving the scalability and computational efficiency for representation learning is also crucial (Killick et al., 2012; Gallagher et al., 2021), especially for handling large and weighted graphs. While our framework demonstrates the ability in change point detection, the development of more sophisticated architectures can enhance the model's capacity on other meaningful tasks (Handcock et al., 2007; Kolar et al., 2010; Yu et al., 2021; Madrid Padilla et al., 2023).

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

430 # 7 Appendix

431 ## 7.1 Updating $\boldsymbol{\mu}$ and $\phi$

In this section, we derive the updates for prior parameter $\boldsymbol{\mu} \in \mathbb{R}^{T \times d}$ and graph decoder parameter $\phi$. Denote the objective function in Equation (4) as $\mathcal{L}(\phi, \boldsymbol{\mu})$ and denote the set of parameters $\{\phi, \boldsymbol{\mu}\}$ as $\boldsymbol{\theta}$. We first calculate the gradient of the log-likelihood $l(\boldsymbol{\theta})$ in $\mathcal{L}(\phi, \boldsymbol{\mu})$ with respect to $\boldsymbol{\theta}$:

$$
\begin{aligned}
\nabla_{\boldsymbol{\theta}} \, l(\boldsymbol{\theta}) &= \nabla_{\boldsymbol{\theta}} \sum_{t=1}^{T} \log P(\boldsymbol{y}^t) \\
&= \sum_{t=1}^{T} \frac{1}{P(\boldsymbol{y}^t)} \nabla_{\boldsymbol{\theta}} P(\boldsymbol{y}^t) \\
&= \sum_{t=1}^{T} \frac{1}{P(\boldsymbol{y}^t)} \nabla_{\boldsymbol{\theta}} \int P(\boldsymbol{y}^t, \boldsymbol{z}^t) d\boldsymbol{z}^t \\
&= \sum_{t=1}^{T} \frac{1}{P(\boldsymbol{y}^t)} \int P(\boldsymbol{y}^t, \boldsymbol{z}^t) \Big[ \nabla_{\boldsymbol{\theta}} \log P(\boldsymbol{y}^t, \boldsymbol{z}^t) \Big] d\boldsymbol{z}^t \\
&= \sum_{t=1}^{T} \int \frac{P(\boldsymbol{y}^t, \boldsymbol{z}^t)}{P(\boldsymbol{y}^t)} \Big[ \nabla_{\boldsymbol{\theta}} \log P(\boldsymbol{y}^t, \boldsymbol{z}^t) \Big] d\boldsymbol{z}^t \\
&= \sum_{t=1}^{T} \int P(\boldsymbol{z}^t | \boldsymbol{y}^t) \Big[ \nabla_{\boldsymbol{\theta}} \log P(\boldsymbol{y}^t, \boldsymbol{z}^t) \Big] d\boldsymbol{z}^t \\
&= \sum_{t=1}^{T} \mathbb{E}_{P(\boldsymbol{z}^t | \boldsymbol{y}^t)} \Big( \nabla_{\boldsymbol{\theta}} \log \Big[ P(\boldsymbol{y}^t | \boldsymbol{z}^t) P(\boldsymbol{z}^t) \Big] \Big) \\
&= \sum_{t=1}^{T} \mathbb{E}_{P(\boldsymbol{z}^t | \boldsymbol{y}^t)} \Big( \nabla_{\boldsymbol{\theta}} \log P(\boldsymbol{y}^t | \boldsymbol{z}^t) \Big) + \sum_{t=1}^{T} \mathbb{E}_{P(\boldsymbol{z}^t | \boldsymbol{y}^t)} \Big( \nabla_{\boldsymbol{\theta}} \log P(\boldsymbol{z}^t) \Big).
\end{aligned}
$$

Note that the expectation in the gradient is now with respect to the posterior distribution $P(\boldsymbol{z}^t | \boldsymbol{y}^t) \propto P(\boldsymbol{y}^t | \boldsymbol{z}^t) \times P(\boldsymbol{z}^t)$. Furthermore, the gradient of $\mathcal{L}(\phi, \boldsymbol{\mu})$ with respect to the prior parameter $\boldsymbol{\mu}^t \in \mathbb{R}^d$ at a specific time point $t$ is

$$
\begin{aligned}
\nabla_{\boldsymbol{\mu}^t} \, \mathcal{L}(\phi, \boldsymbol{\mu}) &= -\mathbb{E}_{P(\boldsymbol{z}^t | \boldsymbol{y}^t)} \Big( \nabla_{\boldsymbol{\mu}^t} \log P(\boldsymbol{z}^t) \Big) + \kappa(\boldsymbol{\mu}^t - \boldsymbol{\nu}^t + \boldsymbol{w}^t) \\
&= -\mathbb{E}_{P(\boldsymbol{z}^t | \boldsymbol{y}^t)}(\boldsymbol{z}^t - \boldsymbol{\mu}^t) + \kappa(\boldsymbol{\mu}^t - \boldsymbol{\nu}^t + \boldsymbol{w}^t).
\end{aligned}
$$

Setting the gradient $\nabla_{\boldsymbol{\mu}^t} \, \mathcal{L}(\phi, \boldsymbol{\mu})$ to zeros and solve for $\boldsymbol{\mu}^t$, we have

$$
\begin{aligned}
\mathbf{0} &= -\mathbb{E}_{P(\boldsymbol{z}^t | \boldsymbol{y}^t)}(\boldsymbol{z}^t) + (1 + \kappa)\boldsymbol{\mu}^t - \kappa(\boldsymbol{\nu}^t - \boldsymbol{w}^t) \\
(1 + \kappa)\boldsymbol{\mu}^t &= \mathbb{E}_{P(\boldsymbol{z}^t | \boldsymbol{y}^t)}(\boldsymbol{z}^t) + \kappa(\boldsymbol{\nu}^t - \boldsymbol{w}^t) \\
\boldsymbol{\mu}^t &= \frac{1}{1 + \kappa} \mathbb{E}_{P(\boldsymbol{z}^t | \boldsymbol{y}^t)}(\boldsymbol{z}^t) + \frac{\kappa}{1 + \kappa}(\boldsymbol{\nu}^t - \boldsymbol{w}^t).
\end{aligned}
$$

Evidently, the gradient of $\mathcal{L}(\phi, \boldsymbol{\mu})$ with respect to the graph decoder parameter $\phi$ is

$$
\nabla_{\phi} \, \mathcal{L}(\phi, \boldsymbol{\mu}) = -\sum_{t=1}^{T} \mathbb{E}_{P(\boldsymbol{z}^t | \boldsymbol{y}^t)} \Big( \nabla_{\phi} \log P(\boldsymbol{y}^t | \boldsymbol{z}^t) \Big).
$$

432 ## 7.2 Langevin Dynamics

Calculating the solution in (7) and the gradient in (8) requires evaluating the conditional expectations under the posterior distribution $P(\boldsymbol{z}^t | \boldsymbol{y}^t) \propto P(\boldsymbol{y}^t | \boldsymbol{z}^t) \times P(\boldsymbol{z}^t)$. In this section, we discuss the Langevin Dynamics

to sample $\boldsymbol{z}^t \in \mathbb{R}^d$ from the posterior distribution $P(\boldsymbol{z}^t|\boldsymbol{y}^t)$ that is conditional on the observed network $\boldsymbol{y}^t \in \{0,1\}^{n \times n}$. The Langevin Dynamics, a short run MCMC, is achieved by iterating the following:

$$
\begin{aligned}
\boldsymbol{z}^t_{\tau+1} &= \boldsymbol{z}^t_\tau + \delta\big[\nabla_{\boldsymbol{z}^t} \log P(\boldsymbol{z}^t|\boldsymbol{y}^t)\big] + \sqrt{2\delta}\boldsymbol{\epsilon} \\
&= \boldsymbol{z}^t_\tau + \delta\big[\nabla_{\boldsymbol{z}^t} \log P(\boldsymbol{y}^t|\boldsymbol{z}^t) + \nabla_{\boldsymbol{z}^t} \log P(\boldsymbol{z}^t) - \nabla_{\boldsymbol{z}^t} \log P(\boldsymbol{y}^t)\big] + \sqrt{2\delta}\boldsymbol{\epsilon} \\
&= \boldsymbol{z}^t_\tau + \delta\big[\nabla_{\boldsymbol{z}^t} \log P(\boldsymbol{y}^t|\boldsymbol{z}^t) - (\boldsymbol{z}^t_\tau - \boldsymbol{\mu}^t)\big] + \sqrt{2\delta}\boldsymbol{\epsilon}
\end{aligned}
$$

where $\tau$ is the time step and $\delta$ is the step size of the Langevin Dynamics. The error term $\boldsymbol{\epsilon} \sim \mathcal{N}(\boldsymbol{0}, \boldsymbol{I}_d)$ serves as a random perturbation to the sampling process. The gradient of the graph decoder $P(\boldsymbol{y}^t|\boldsymbol{z}^t)$ with respect to the latent variable $\boldsymbol{z}^t$ can be calculated efficiently through back-propagation. Essentially, we use MCMC samples to approximate the conditional expectation $\mathbb{E}_{P(\boldsymbol{z}^t|\boldsymbol{y}^t)}(\cdot)$ in the solution (7) and the gradient (8).

### 7.3   Group Lasso for Updating $\beta$

In this section, we present the derivation to update $\boldsymbol{\beta}$ in Proposition 2, which is equivalent to solving a Group Lasso problem Yuan & Lin (2006). We adapt the derivation from Bleakley & Vert (2011) for our proposed ADMM algorithm. Denote the objective function in (5) as $\mathcal{L}(\boldsymbol{\gamma}, \boldsymbol{\beta})$. When $\boldsymbol{\beta}_{t,\cdot} \neq \boldsymbol{0}$, the gradient of $\mathcal{L}(\boldsymbol{\gamma}, \boldsymbol{\beta})$ with respect to $\boldsymbol{\beta}_{t,\cdot}$ is

$$
\nabla_{\boldsymbol{\beta}_{t,\cdot}}\mathcal{L}(\boldsymbol{\gamma},\boldsymbol{\beta}) = \lambda\frac{\boldsymbol{\beta}_{t,\cdot}}{\|\boldsymbol{\beta}_{t,\cdot}\|_2} - \kappa\boldsymbol{X}_{\cdot,t}^\top(\boldsymbol{\mu}_{(a+1)} + \boldsymbol{w}_{(a)} - \boldsymbol{1}_{T,1}\boldsymbol{\gamma} - \boldsymbol{X}_{\cdot,t}\boldsymbol{\beta}_{t,\cdot} - \boldsymbol{X}_{\cdot,-t}\boldsymbol{\beta}_{-t,\cdot})
$$

where $\boldsymbol{X}_{\cdot,t} \in \mathbb{R}^{T \times 1}$ is the $t$-th column of matrix $\boldsymbol{X} \in \mathbb{R}^{T \times (T-1)}$ and $\boldsymbol{\beta}_{t,\cdot} \in \mathbb{R}^{1 \times d}$ is the $t$-th row of matrix $\boldsymbol{\beta} \in \mathbb{R}^{(T-1) \times d}$. Moreover, we denote $\boldsymbol{\beta}_{-t,\cdot} \in \mathbb{R}^{(T-1) \times p}$ as the matrix obtained by replacing the $t$-th row of matrix $\boldsymbol{\beta}$ with a zero vector, and $\boldsymbol{X}_{\cdot,-t} \in \mathbb{R}^{T \times (T-1)}$ is denoted similarly.

Setting the above gradient to zeros, we have

$$
\boldsymbol{\beta}_{t,\cdot} = (\kappa\boldsymbol{X}_{\cdot,t}^\top\boldsymbol{X}_{\cdot,t} + \frac{\lambda}{\|\boldsymbol{\beta}_{t,\cdot}\|_2})^{-1}\boldsymbol{b}_t \tag{14}
$$

where

$$
\boldsymbol{b}_t = \kappa\boldsymbol{X}_{\cdot,t}^\top(\boldsymbol{\mu}_{(a+1)} + \boldsymbol{w}_{(a)} - \boldsymbol{1}_{T,1}\boldsymbol{\gamma} - \boldsymbol{X}_{\cdot,-t}\boldsymbol{\beta}_{-t,\cdot}) \in \mathbb{R}^{1 \times d}.
$$

Calculating the Euclidean norm of (14) on both sides and rearrange the terms, we have

$$
\|\boldsymbol{\beta}_{t,\cdot}\|_2 = (\kappa\boldsymbol{X}_{\cdot,t}^\top\boldsymbol{X}_{\cdot,t})^{-1}(\|\boldsymbol{b}_t\|_2 - \lambda).
$$

Plugging $\|\boldsymbol{\beta}_{t,\cdot}\|_2$ into (14) for substitution, the solution of $\boldsymbol{\beta}_{t,\cdot}$ is arrived at

$$
\boldsymbol{\beta}_{t,\cdot} = \frac{1}{\kappa\boldsymbol{X}_{\cdot,t}^\top\boldsymbol{X}_{\cdot,t}}(1 - \frac{\lambda}{\|\boldsymbol{b}_t\|_2})\boldsymbol{b}_t.
$$

Moreover, when $\boldsymbol{\beta}_{t,\cdot} = \boldsymbol{0}$, the subgradient $\boldsymbol{v}$ of $\|\boldsymbol{\beta}_{t,\cdot}\|_2$ needs to satisfy that $\|\boldsymbol{v}\|_2 \leq 1$. Because

$$
\boldsymbol{0} \in \lambda\boldsymbol{v} - \kappa\boldsymbol{X}_{\cdot,t}^\top(\boldsymbol{\mu}_{(a+1)} + \boldsymbol{w}_{(a)} - \boldsymbol{1}_{T,1}\boldsymbol{\gamma} - \boldsymbol{X}_{\cdot,-t}\boldsymbol{\beta}_{-t,\cdot}),
$$

we obtain the condition that $\boldsymbol{\beta}_{t,\cdot}$ becomes $\boldsymbol{0}$ when $\|\boldsymbol{b}_t\|_2 \leq \lambda$. Therefore, we can iteratively apply the following to update $\boldsymbol{\beta}_{t,\cdot}$ for each block $t = 1, \ldots, T-1$:

$$
\boldsymbol{\beta}_{t,\cdot} \leftarrow \frac{1}{\kappa\boldsymbol{X}_{\cdot,t}^\top\boldsymbol{X}_{\cdot,t}}\left(1 - \frac{\lambda}{\|\boldsymbol{b}_t\|_2}\right)_+ \boldsymbol{b}_t
$$

where $(\cdot)_+ = \max(\cdot, 0)$.

### 7.4 Practical Guidelines

#### 7.4.1 ADMM Implementation

In this section, we provide practical guidelines for the proposed framework and the Alternating Direction Method of Multipliers (ADMM) algorithm. For Langevin Dynamic sampling, we set $\delta = 0.5$, and we draw $s = 200$ samples for each time point $t$. To detect change points using the data-driven threshold in (13), we let the tuning parameter $\lambda = \{10, 20, 50, 100\}$. To detect change points using the localizing method with Gamma distribution in (11), we let the tuning parameter $\lambda = \{5, 10, 20, 50\}$. For each $\lambda$, we run $A = 50$ iterations of ADMM. Within each ADMM iteration, we run $B = 20$ iterations of gradient descent with Adam optimizer for the graph decoder and $D = 20$ iterations of block coordinate descent for Group Lasso.

Since the proposed generative model is a probability distribution for the observed network data, in this work we stop ADMM learning with the following stopping criteria:

$$\left| \frac{l(\boldsymbol{\phi}_{(a+1)}, \boldsymbol{\mu}_{(a+1)}) - l(\boldsymbol{\phi}_{(a)}, \boldsymbol{\mu}_{(a)})}{l(\boldsymbol{\phi}_{(a)}, \boldsymbol{\mu}_{(a)})} \right| \leq \epsilon_{\text{tol}}. \tag{15}$$

The log-likelihood $l(\boldsymbol{\phi}, \boldsymbol{\mu})$ is approximated by sampling from the prior distribution $p(\boldsymbol{z}^t)$, as described in Section 4.2. Hence, we stop the ADMM procedure until the above criteria is satisfied for $a'$ consecutive iterations. In Section 5, we set $\epsilon_{\text{tol}} = 10^{-5}$ and $a' = 5$.

Here we briefly elaborate on the computational aspect of the approximation of the log-likelihood. To calculate the product of edge probabilities for the conditional distribution $P(\boldsymbol{y}^t | \boldsymbol{z}^t)$, we have the following:

$$\sum_{t=1}^{T} \log P(\boldsymbol{y}^t) = \sum_{t=1}^{T} \log \int P(\boldsymbol{y}^t | \boldsymbol{z}^t) P(\boldsymbol{z}^t) d\boldsymbol{z}^t$$

$$= \sum_{t=1}^{T} \log \mathbb{E}_{P(\boldsymbol{z}^t)} \Big[ \prod_{(i,j) \in \mathbb{Y}} P(\boldsymbol{y}_{ij}^t | \boldsymbol{z}^t) \Big]$$

$$\approx \sum_{t=1}^{T} \log \Big[ \frac{1}{s} \sum_{u=1}^{s} [\prod_{(i,j) \in \mathbb{Y}} P(\boldsymbol{y}_{ij}^t | \boldsymbol{z}_u^t)] \Big]$$

$$= \sum_{t=1}^{T} \log \Big[ \frac{1}{s} \sum_{u=1}^{s} \exp\{ \sum_{(i,j) \in \mathbb{Y}} \log[P(\boldsymbol{y}_{ij}^t | \boldsymbol{z}_u^t)] \} \Big]$$

$$= \sum_{t=1}^{T} \Big\{ -\log s + \log \Big[ \exp C^t \sum_{u=1}^{s} \exp\{ \sum_{(i,j) \in \mathbb{Y}} \log[P(\boldsymbol{y}_{ij}^t | \boldsymbol{z}_u^t)] - C^t \} \Big] \Big\}$$

$$= \sum_{t=1}^{T} \Big\{ C^t + \log \Big[ \sum_{u=1}^{s} \exp\{ \sum_{(i,j) \in \mathbb{Y}} \log[P(\boldsymbol{y}_{ij}^t | \boldsymbol{z}_u^t)] - C^t \} \Big] \Big\} - T \log s$$

where $C^t \in \mathbb{R}$ is the maximum value of $\sum_{(i,j) \in \mathbb{Y}} \log[P(\boldsymbol{y}_{ij}^t | \boldsymbol{z}_u^t)]$ over $m$ samples but within a time point $t$.

We also update the penalty parameter $\kappa$ to improve convergence and to reduce reliance on its initialization. In particular, after the $a$-th ADMM iteration, we calculate the respective primal and dual residuals:

$$r_{\text{primal}}^{(a)} = \sqrt{\frac{1}{T \times d} \sum_{t=1}^{T} \|\boldsymbol{\mu}_{(a)}^t - \boldsymbol{\nu}_{(a)}^t\|_2^2} \quad \text{and} \quad r_{\text{dual}}^{(a)} = \sqrt{\frac{1}{T \times d} \sum_{t=1}^{T} \|\boldsymbol{\nu}_{(a)}^t - \boldsymbol{\nu}_{(a-1)}^t\|_2^2}.$$

Throughout, we initialize the penalty parameter $\kappa = 10$. We jointly update the penalty parameter $\kappa$ and the scaled dual variable $\boldsymbol{w}$ as in Boyd et al. (2011) with the following conditions:

$$\kappa_{(a+1)} = 2\kappa_{(a)}, \quad \boldsymbol{w}_{(a+1)} = \frac{1}{2}\boldsymbol{w}_{(a)}, \quad \text{if } r_{\text{primal}}^{(a)} > 10 \times r_{\text{dual}}^{(a)},$$

$$\kappa_{(a+1)} = \frac{1}{2}\kappa_{(a)}, \quad \boldsymbol{w}_{(a+1)} = 2\boldsymbol{w}_{(a)}, \quad \text{if } r_{\text{dual}}^{(a)} > 10 \times r_{\text{primal}}^{(a)}.$$

### 7.4.2 Post-Processing

Since neural networks may be over-fitted for a statistical model in change point detection, we track the following Coefficient of Variation as a signal-to-noise ratio when we learn the model parameter with the full data:

$$\text{Coefficient of Variation} = \frac{\text{mean}(\Delta\hat{\boldsymbol{\mu}})}{\text{sd}(\Delta\hat{\boldsymbol{\mu}})}.$$

We choose the learned parameter $\hat{\boldsymbol{\mu}}$ with the largest Coefficient of Variation as final output.

By convention, we also implement two post-processing steps to finalize the detected change points. When the gap between two consecutive change points is small or $\hat{C}_k - \hat{C}_{k-1} < \epsilon_{\text{spc}}$, we preserve the detected change point with greater $\Delta\hat{\boldsymbol{\zeta}}$ value to prevent clusters of nearby change points. Moreover, as the endpoints of a time span are usually not of interest, we remove the $\hat{C}_k$ smaller than a threshold $\epsilon_{\text{end}}$ and the $\hat{C}_k$ greater than $T - \epsilon_{\text{end}}$. In Section 5, we set $\epsilon_{\text{spc}} = 5$ and $\epsilon_{\text{end}} = 5$.

