# OpenReview forum: "Generative Model for Change Point Detection in Dynamic Graphs"
_TMLR — Rejected by TMLR_

### Review · Reviewer_dn3L · 2024-08-13

**Summary Of Contributions:**

The article studies the problem of Change-Point Detection in the context of graphs. That is to say, we observe a set of $T$ graphs (indexed by $t$ which is assumed to correspond to a certain temporal notion) and we would like to detect if and when the generative process underlying each graph changes.

To this end, the authors propose a generative model, where a latent vector $\mathbf{z}\in\mathbb{R}^d$ is assumed to exist, which in turn generates the graph through an inhomogeneous Bernoulli model. In particular, an edge between nodes $i$ and $j$ exists with probability given by $g(\mathbf{U}\mathbf{U}^\top)$ (for the undirected case), where $\mathbf{U}=\text{MLP}(\mathbf{z})$ is a multi-layer perceptron (which maps the input $d$-dimensional vector to a $n\times k$ matrix, being $n$ the number of nodes) and $g(\cdot)$ is a point-to-point sigmoid.

Instead of trying to find (fit) $\mathbf{z}$ for each $t$, the paper studies an approach very similar to a Variational Autoencoder (VAE), where $\mathbf{z}$ is random and a certain prior distribution is assumed (in this case an uncorrelated Gaussian vector). The problem then becomes how to fit the mean of this distribution $\boldsymbol{\mu}$ for each $t$, and a change-point will be flagged if the change between time-steps is high enough. Actually, two methods are proposed, although the second one, a relatively simple heuristic, proves to work best in practice.

I believe that the main contribution is presenting an objective function specially tailored for CPD, and a method to solve the resulting optimization problem. Instead of using ELBO as in most VAEs, the authors considered the log-likelihood plus a regularization term that penalizes differences between consecutive $\boldsymbol{\mu}$ (instead of the KL-divergence of ELBO). Experimental results including synthetic as well as real-world data confirm the competitiveness of the proposal w.r.t. other methods.

**Audience:**

Yes

**Broader Impact Concerns:**

None.

**Claims And Evidence:**

No

**Requested Changes:**

The changes I request are the following. I've discussed the reasons behind these changes the previous section.

1. A thorough comparison to (Graph) VAE methods is in order.
2. Section 2.1 (and consequently 2.2) should be revised to avoid confusing notation.
3. Include a discussion on the computation cost of the proposed method.
4. Include a discussion on the limitations of the proposal, including why it fails to detect some fairly evident change-point in the real-world dataset, and the expressivity of the proposed generative method.
5. The source code should be made available. This would allow further understanding of the method, as well as enable other researchers to compare the proposed method to others.

**Strengths And Weaknesses:**

As I mentioned before, I believe that the main contribution and thus biggest strength of the paper is the proposal of the cost in Equation (1), and the method to solve the resulting optimization problem.

However, the paper has several shortcomings. Maybe most importantly is the lack of discussion regarding Variational Autoencoders. The decoder is extremely similar to to the classic GraphVAE [1], in that they also use a graph-level embedding with a Gaussian prior. Three important differences exists with the method proposed here:
1. Where GraphVAE generates the graph directly from a MLP, here the MLP produces an intermediate per-node representation which then produces the connection probability through a dot product.
2. As I mentioned before, the objective function is not ELBO.
3. The encoder (which in GraphVAE is a GNN followed by a pooling) is not present.

I believe that the paper would significantly benefit from a discussion that compares GraphVAE (or any of the similar methods that have been proposed since, including a recent article bearing on CPD [2]) with the method presented here. For instance, what is the benefit of not using an encoder? Using a GNN as in GraphVAE provides the advantage of using simple backpropagation to optimize. The method proposed here is not computationally simple, or at least no discussion regarding its computational cost was included, so the advantage is not clear.

Regarding point 1 above, and the expressive power of the generative model (which is highlighted in the abstract), I believe this is somewhat of an overstatement. Simulations to assess the expressive power of the model are missing (only to evaluate its usefulness for CPD). Furthermore, the expression $\mathbf{U}\mathbf{U}^\top$ means that only positive semi-definite probability matrices can be generated, thus negating the possibility of generating heterophily on the resulting graphs.

A further weakness lies on the presentation, particularly the notation defined throughout section 2.1. For instance, symbol $\mathbb{Y}$ is used to design the set of pairs of nodes that are connected, and then $2^{\mathbb{Y}}$ appears. I would suggest to carefully read this section to avoid this kind of confusing definitions.

Regarding the experiments, the real-world data has been thoroughly studied. I personally particularly like figure 8 of https://arxiv.org/abs/1403.0989, which marks several events of both MIT and Enron. Regarding the latter, I found it surprising and somewhat disappointing that the proposed method fails to detect the launch of Enron Online and the assumption of Stephen Cooper as CEO events, which are easily detected by simply counting the number of edges in the graphs (both of which are detected by gSeg and kerSeg). Regarding the former, none of the methods detected the beginning of the semester, which is also surprising.

In any case, I would have liked a honest discussion regarding the limitations of the proposed method. For instance, nodes have to be the same for all time-steps. How did you proceed in the real-world datasets? For instance, in the Enron case not everyone sends or receives mails during all weeks.

[1] Simonovsky, M., Komodakis, N. (2018). GraphVAE: Towards Generation of Small Graphs Using Variational Autoencoders. In: Kůrková, V., Manolopoulos, Y., Hammer, B., Iliadis, L., Maglogiannis, I. (eds) Artificial Neural Networks and Machine Learning – ICANN 2018. ICANN 2018. Lecture Notes in Computer Science(), vol 11139. Springer, Cham. https://doi.org/10.1007/978-3-030-01418-6_41
[2] X. Zhang et al., "VGGM: Variational Graph Gaussian Mixture Model for Unsupervised Change Point Detection in Dynamic Networks," in IEEE Transactions on Information Forensics and Security, vol. 19, pp. 4272-4284, 2024, doi: 10.1109/TIFS.2024.3377548.

---

> ### Author Response · Authors · 2024-10-13
> **Response to Review Comments**
>
> We are extremely grateful for the constructive comments and especially for pointing out the discussion with GraphVAE and VGGM. We have also uploaded an updated manuscript based on the requested changes.
>
> $\bf{Q1}$: We have modified the manuscript for this comment from Line 46 to Line 53 on Page 2, from Line 82 to Line 85 on Page 3, and from Line 103 to Line 111 on Page 4. Below are our responses to this comment.
>
> The GraphVAE [1] encodes each graph $\bf{G}$ to a latent $\bf{z}$ that follows the approximate posterior $q(\bf{z}|\bf{G})$, and it regularizes the latent space to be close to a pre-defined prior $p(\bf{z}) = N(\bf{z};\bf{0},\bf{I})$ through $KL(q(\bf{z}|\bf{G}) || p(\bf{z}) )$. The parameters in GraphVAE are learned by maximizing the Evidence Lower Bound (ELBO) for the log-likelihood of the marginal distribution $p(\bf{G})$, where the KL term is derived from the ELBO and introduced as regularization to the latent space for $\bf{z}$.
>
> In our proposed framework for a time series of graphs, for a network $\bf{y}^t$ at a time point $t$, we assume the latent $\bf{z}^t$ follows a prior $p(\bf{z}^t) = N(\bf{z}^t;\bf{\mu}^t,\bf{I})$, where the mean $\bf{\mu}^t$ is estimated from the network data. Also, we let the priors $p(\bf{z}^t)$ differed by time $t$. In other words, for a sequence of $T$ networks $\bf{y}^1,\dots,\bf{y}^T$, we have a total of $T$ priors $p(\bf{z}^t)$ to be estimated.
>
> [Comparison] In our proposed objective function and learning method, we maximize an approximation to the log-likelihood of the distribution for the time series of graphs, while penalizing the sum of sequential differences between the prior parameters or $\sum_{t=1}^{T-1} ||\bf{\mu}^{t+1} - \bf{\mu}^{t}||_2$, which is a Group Fused Lasso penalty. When one uses the ELBO of the log-likelihood of the $p(\bf{G})$ as objective function, the approximate posterior $q(\bf{z}|\bf{G})$ from the $KL(q(\bf{z}|\bf{G}) || p(\bf{z}))$ term requires the framework to learn an encoder [1,2,3]. As we do not use the ELBO as the objective function, an encoder is not required for our proposed framework. Essentially, the VAE learns an encoder for $q(\bf{z}|\bf{G})$ and decoder for $p(\bf{G}|\bf{z})$, while our framework learns the prior $p(\bf{z}^t)$ and a decoder for $p(\bf{y}^t|\bf{z}^t)$. Also, as we learn the priors $p(\bf{z}^t) = N(\bf{z}^t; \bf{\mu}^t,\bf{I})$ from data, we do not enforce the priors $p(\bf{z}^t)$ to have $\bf{\mu}^t = \bf{0}$ as in a VAE framework. In contrast, VAE specifies a zero-mean Gaussian prior $p(\bf{z}) = N(\bf{z};\bf{0},\bf{I})$ and encourages the approximate posterior $p(\bf{z}|\bf{G})$ to be close to the zero-mean prior  $p(\bf{z})$. In our work, we specify the action of detection as Group Fused Lasso penalty in the objective function, so that the learned priors can incorporate the structural changes between consecutive graphs into the latent space. Without the penalty term, the learned $\bf{z}^t$ may not be helpful to a change point detection task, even if the $\bf{z}^t$ is a good representation of the graph $\bf{y}^t$ for other tasks.
>
> In our proposed model architecture, we use Multi-layer Perceptrons (MLP) to produce $\bf{U}^t, \bf{V}^t \in \mathbb{R}^{n \times k}$ instead of directly outputting the graph $\bf{y}^t \in \\{0,1\\}^{n \times n}$, to reduce the number of parameters in the neural networks. When the number of nodes $n$ is large, the number of parameters in the MLP-based decoder can grow very fast, if it directly outputs the graph $\bf{y}^t$. Moreover, graphs can be sparse, and having MLP directly to output the graphs can be over-parameterized for a model. As the graph generation through $\bf{U}^t {\bf{V}^t}^\top$ for directed graphs and $\bf{U}^t {\bf{U}^t}^\top$ for undirected graphs is common in the literature [3,4,5], we focus on this setting for graphs where nodes with homophily tend to form edges. We will consider other types of generation to tackle the heterophily problem in a future work.

---

> > ### Author Response · Authors · 2024-10-13
> > **Response to Review Comments**
> >
> > [Limitation] Since we do not need to learn an approximate posterior $q(\bf{z}^t|\bf{y}^t)$ to regularize the latent space with a pre-defined prior $p(\bf{z}^t) = N(\bf{z}^t;\bf{0},\bf{I})$, we learn the prior parameters $\bf{\mu}^t$ of $p(\bf{z}^t) = N(\bf{z}^t;\bf{\mu}^t,\bf{I})$ from data as empirical Bayes. However, the trade-off arises from the computational aspect. More specifically, we have to implement Langevin Dynamics to draw MCMC samples from the posterior $p(\bf{z}^t|\bf{y}^t) \propto p(\bf{y}^t|\bf{z}^t) p(\bf{z}^t)$ to estimate the prior parameters, and we need to implement ADMM to deal with the Group Lasso penalty, which does not work well with gradient descent type of methods. In general, using Bayesian inference to learn the prior $p(\bf{z}^t)$ and to draw samples from the posterior $p(\bf{z}^t|\bf{y}^t)$ is often computationally intensive. We plan to improve the computational efficiency and to handle varied node set, by leveraging the ideas from GraphVAE [1] and VGGM [2], when we extend our framework to dynamic weighted networks in a future work.
> >
> > $\bf{Q2}$: For a node set $N = \\{1,2,\cdots,i,j,\cdots, n\\}$, we use the notation $(i,j) \in \mathbb{Y} \subseteq N \times N$ to denote the possible pairs that can be formed by $n$ nodes. The pair $(i,j)$ can be either connected or not. We use the notation $\mathcal{Y} \subseteq 2^\mathbb{Y}$ to denote the sample space for an $n$ by $n$ adjacency matrix or network $\bf{y} \in \mathcal{Y}$, where a dyad $\bf{y}_{ij} \in \\{0,1\\}$ can take either $0$ or $1$ for the $(i,j)$ pair.
> >
> > $\bf{Q3}$: We have modified the manuscript for this comment from Line 141 to Line 146 on Page 6. Below is the corresponding text from the updated manuscript.
> >
> > "The complexity of the proposed algorithm is at least of order $O\big(A(Tsl + BT + D(T-1))\big)$ with additional gradient calculation for neural networks in the sub-routines. Specifically, for each of the $A$ iterations of ADMM, we update the prior parameters $\bf{\mu}^t$ for all $T$ time points, and each update involves $l$ steps of MCMC for $s$ samples. Then we calculate the gradients for neural networks over the $T$ time points and run $B$ iterations of Adam optimizer. Lastly, we run $D$ iterations of block coordinate descent for the $T-1$ sequential differences."
> >
> > $\bf{Q4}$: We have modified the manuscript for this comment from Line 4 to Line 6, from Line 9 to Line 11 on Page 1, and from Line 43 to Line 46 on Page 2. Below are our responses to this comment.
> >
> > Change point detection is inherently an unsupervised problem. The goal is to identify time points where significant changes occur, without prior knowledge of the timing of these changes. Often, there is no widely accepted benchmark that outlines which real-world events should be considered as true changes. Though we have a list of real-world events from public news on Enron, the challenge lies in determining which of these events should be classified as true change points, as their impact toward graph structures can be different. In 2001, Enron underwent a multitude of major and overlapping incidents, making it difficult to associate the changes in graph structures with specific real-world events.
> >
> > Figure 6 depicts the change magnitude $\Delta \hat{\bf{\zeta}}$ over time. The assumption of Stephen Cooper as CEO and other nearby by events align with the small spikes that did not exceed the red threshold toward the end in Figure 6. The event of Stephen Cooper as CEO may not stand out as a strong change point, especially after the large spike in $\Delta \hat{\bf{\zeta}}$ that aligned with Enron filed for bankruptcy. However, one can lower the threshold to adjust the sensitivity of our detection method if needed. Also, the launch of Enron Online in Nov 1999 is not included in the data (June 2000 - May 2002) we applied our method. For employee $i$ that did not send an email to employee $j$ at time $t$, we have $\bf{y}_{ij}^t = 0$.
> >
> > [Application] We focus on using a generative model for a change point detection task. Recently, generative models have been used in a wide range of application, such as chatbot via text generation, scene synthesis via video generation, and drug discovery via graph generation. In this work, we explore how generative model can assist a change point detection task, as there are other detection methods without using a generative model. We have tuned down the statement in the abstract to avoid confusion, as we are using generative model to detect change points in time series of graphs, instead of comparing the fidelity of the generated graphs with other methods for generation tasks.

---

> > > ### Author Response · Authors · 2024-10-13
> > > **Response to Review Comments**
> > >
> > > $\bf{Q5}$: We will publish the source code online.
> > >
> > > [1] Simonovsky, M., \& Komodakis, N., "GraphVAE: Towards generation of small graphs using variational autoencoders." Artificial Neural Networks and Machine Learning – ICANN, 2018.
> > >
> > > [2] Zhang, X., Jiao, P., Gao, M., Li, T., Wu, Y., Wu, H., \& Zhao, Z., "VGGM: Variational Graph Gaussian Mixture Model for Unsupervised Change Point Detection in Dynamic Networks." IEEE Transactions on Information Forensics and Security, 2024.
> > >
> > > [3] Kipf, T. N., \& Welling, M., "Variational Graph Auto-Encoders". NeurIPS Workshop on Bayesian Deep Learning - NeurIPS BDL, 2016.
> > >
> > > [4] Hamilton, W., Ying, Z., \& Leskovec, J., "Inductive Representation Learning on Large Graphs." Advances in neural information processing systems, 2017.
> > >
> > > [5] Pan, S., Hu, R., Long, G., Jiang, J., Yao, L., \& Zhang, C., "Adversarially Regularized Graph Autoencoder for Graph Embedding". Proceedings of the 27th International Joint Conference on Artificial Intelligence, 2018.

---

> ### Comment · Reviewer_dn3L · 2024-10-25
> **Regarding notation**
>
> Thanks for the answers. If I may, I insist on the notation. If $N=\{1,...,n\}$, then $N\times N$ is the possible pairs that can be formed by $n$ nodes. The notation in use seems to imply that $\mathbb{Y}$ are the actual edges that do exist. What would be the sense of defining "potential relations" that do not include all possible pairs of nodes in a random model such as the one you are using?
>
> Note furthermore that "network, graph or adjacency matrix" are not the same thing and should not be denoted by simply $\mathbf{y}$. An adjacency matrix is a representation of a graph, not a graph itself. If $\mathbf{y}$ will denote an adjacency matrix, then $\mathbf{y}\in {0,1}^{|N|\times |N|}$ (or $\mathbf{y}\in {0,1}^{n \times n}$). I'm not sure what the power of 2 of a set is.
>
> Edit: not sure why the braces are not shown in the answer... Sorry for that!

---

> > ### Author Response · Authors · 2024-10-25
> > **Response to Review Comments**
> >
> > We are very grateful for the reviewer’s comments regarding the clarity of our notation. We agree that the original presentation could be improved for readability. Accordingly, we have updated Lines 71 to 74 in the revised manuscript to enhance clarity, as shown in the updated text below:
> >
> > "For a node set $N = \\{1,2,\cdots,n\\}$, we use an adjacency matrix $y \in \\{0,1\\}^{n \times n}$ to represent a graph. We denote the set of all possible node pairs as $\mathbb{Y} = N \times N$. In the adjacency matrix, $y_{ij} = 1$ indicates an edge between nodes $i$ and $j$, while $y_\{ij\} = 0$ indicates no edge. The relations can be either directed or undirected. The undirected variant has $y_{ij} = y_{ji}$ for all $(i,j) \in \mathbb{Y}$."
> >
> > We believe this revision makes the notation easier to understand, and we appreciate the reviewer’s input in helping us improve this aspect of the paper (the bold font is not showing in the above quote).

---

### Review · Reviewer_kkTr · 2024-08-19

**Summary Of Contributions:**

This paper proposes a generative model that learns a prior distribution for the low-dimensional representation of graph data. This model is then used to detect change points in time series of graphs by analyzing changes in the latent representations.

**Audience:**

Yes

**Claims And Evidence:**

Yes

**Requested Changes:**

See weakness section above.

**Strengths And Weaknesses:**

Strengths:
1. The method is grounded in a rigorous generative graph model.
2. The optimization problem, utilizing group lasso and ADMM, is well-posed and appropriately formulated.

Weaknesses:
1. The paper’s scope is somewhat narrow, focusing solely on detecting change points. Existing literature, such as [1], not only detects change points but also quantifies the magnitude of changes in both the graph and its vertices. It would be interesting to explore whether the current method could be extended to detect vertex-level changes, in addition to identifying the time points of graph changes.

2. There is a lack of theoretical quantification or intuitive explanation on the algorithm design. Sections 3 and 4 employ a complex algorithm, which, while viable, lacks a clear rationale. One could argue that a simpler approach might be equally effective. A discussion on why the authors chose this particular algorithm, such as some theoretical justification, and / or a comparison to simpler methods such as spectral embedding [2], would provide valuable insight.

3. The scalability and computational efficiency of the method need to be addressed. The simulations and real-data experiments are conducted on relatively small graphs and time-steps. While this does not inherently diminish the method’s value, a discussion of the running time is desired.

4. The advantages of the method in simulations are not really significant, except scenario 3 recurrent neural networks. From my perspective, recurrent neural networks are specialized artificial constructs and may be the only scenarios where the designed algorithm performs optimally. In this context, the authors should either provide a clearer explanation of the algorithm’s intuition (as mentioned in point 2) or identify additional simulation scenarios where the algorithm demonstrates significant advantages.

5. The real-data experiments lack ground-truth comparisons, making it difficult to evaluate the effectiveness of different methods.

[1] Shen et al., "Discovering Communication Pattern Shifts in Large-Scale Networks using Encoder Embedding and Vertex Dynamics," IEEE TNSE, 2024.

[2] Gallagher et al., "Spectral embedding for dynamic networks with stability guarantees", NeurIPS, 2021.

---

> ### Author Response · Authors · 2024-10-13
> **Response to Review Comments**
>
> We are extremely grateful for the constructive comments and especially for pointing out the intuition of our proposed framework. We also thank the reviewer for suggesting change detection at the vertex level and for the quantification of real-data experiments. We have also uploaded an updated manuscript based on the requested changes.
>
> $\bf{Q1}$: We have modified the manuscript for this comment from Line 284 to Line 287 on Page 15. Below are our responses to this comment.
>
> In this work, we primarily focus on detecting changes on the graph structures over time. It is possible to detect changes in vertices [1], if we impose a prior distribution to each row of $\bf{U}^t, \bf{V}^t \in \mathbb{R}^{n \times k}$, where $\bf{U}^t$ and $\bf{V}^t$ are considered as the node level representation in our framework. We can penalize the sequential differences of the prior parameters for the nodal representation and detect the changes in vertices, similar to our purposed method for detecting changes in graph structures. For future development, we plan to distinguish between nodal attributes (such as gender and race that may not be effected by the graphs) and other types of node level representation (that could be altered by the graphs over time) to facilitate change detection at vertex level in dynamic graphs.
>
> $\bf{Q2}$: We have modified the manuscript for this comment from Line 103 to Line 111 on Page 4. Below are our responses to this comment.
>
> The intuition of our framework is that we learn the representation for graphs over time, with the action of change detection specified in the objective function of Eq. (1), in the form of Group Fused Lasso (GFL) penalty $\sum_{t=1}^{T-1} ||\bf{\mu}^{t+1} - \bf{\mu}^{t}||_2$ over the prior parameters.  The GFL penalty enforces the multivariate prior parameter $\bf{\mu}^t \in \mathbb{R}^d$ to remain constant between two consecutive change points, while allowing multiple coordinates across the $d$ dimensional differences $\bf{\mu}^t - \bf{\mu}^{t-1}$ to change at the same time $t$. By penalizing the sum of sequential differences, the proposed framework focuses on capturing meaningful structural changes and smoothing out minor variations. Without using the GFL penalty in the objective function, the learned representation $\bf{z}^t$ may not be helpful for change point detection, even if the $\bf{z}^t$ is a good representation of the graph $\bf{y}^t$ for other tasks.
>
> Furthermore, we choose this particular learning procedure because we have to learn the prior distribution $p(\bf{z}^t)$ for the graph level representation $\bf{z}^t$ and to solve a constrained optimization problem. Instead of using an encoder to embed a graph $\bf{y}^t$ to obtain a representation $\bf{z}^t$ as in [2], we infer the parameters $\bf{\mu}^t$ of the prior distribution $p(\bf{z}^t) = N(\bf{z}^t;\bf{\mu}^t,\bf{I})$ for the representation $\bf{z}^t$ as empirical Bayes, with GFL penalty helping with change point detection. The resulting augmented Lagrangian is then solved via ADMM, which is well-studied in terms of both theory and practice [3,4]. In summary, we choose this framework to learn the prior distributions of the graph level representations that are useful for change point detection.
>
> $\bf{Q3}$: We have modified the manuscript for this comment from Line 141 to Line 146 on Page 6. Below is the corresponding text from the updated manuscript.
>
> "The complexity of the proposed algorithm is at least of order $O\big(A(Tsl + BT + D(T-1))\big)$ with additional gradient calculation for neural networks in the sub-routines. Specifically, for each of the $A$ iterations of ADMM, we update the prior parameters $\bf{\mu}^t$ for all $T$ time points, and each update involves $l$ steps of MCMC for $s$ samples. Then we calculate the gradients for neural networks over the $T$ time points and run $B$ iterations of Adam optimizer. Lastly, we run $D$ iterations of block coordinate descent for the $T-1$ sequential differences.''
>
> Bayesian inference to learn the prior $p(\bf{z}^t)$ and to draw samples from the posterior $p(\bf{z}^t|\bf{y}^t)$ is often computationally intensive. We plan to enhance the scalability and computational efficiency by leveraging the ideas from [1,2] when we extend our framework to dynamic weighted networks for a future work.

---

> ### Author Response · Authors · 2024-10-13
> **Response to Review Comments**
>
> $\bf{Q4}$: In Scenario 1 and 2 of the simulation study, the competitor methods use the true network statistics, that generate the artificial data, to detect the change points. The best competitor methods are considered as oracle and they are expected to perform well by `cheating'. In practice, the true network statistics are usually not known to the modeler a prior, which motivates representation learning for change point detection problem in dynamic graphs. Although our proposed method does not outperform the best competitors significantly in Scenario 1, our method is robust to different scenarios and achieves the best results in Scenario 2 and 3. We do not need to specify the network statistics that cause the change in network structure to achieve similar and better performance.
>
> In Scenario 1, we use STERGM with three network statistics: (1) edge count, (2) mutuality, and (3) number of triangles to generate the artificial networks. Then the CPDstergm method with $p=6$ also uses these three network statistics to detect the change points in generated graphs. Though CPDstergm with $p=6$ achieves the best result, when we remove one true network statistic from the CPDstergm method (i.e. $p=4$), the performance of CPDstergm drops significantly. Our method achieves similar performance to the best competitor.
>
> In Scenario 2, we use SBM to generate the simulated networks with block structures, where mutuality serves as an important pattern for the homophily within groups. Then the CPDstergm method with $p=4$ uses both (1) edge count and (2) mutuality to detect change points. Our proposed method, without specifying the features that induce the changes, achieves the best result in this scenario.
>
> In Scenario 3, we use RNN to generate the artificial networks. While no obvious change pattern is visible, the competitor methods which rely on the choice of network statistics cannot produce good performance. Our proposed method, infers the features in latent space that induce the structural changes, achieves the best result in this scenario.
>
> $\bf{Q5}$: We recognize the reviewer’s concern regarding the lack of ground-truth in real-data experiments. Change point detection is inherently an unsupervised problem, as the goal is to identify time points where significant changes occur, without prior knowledge of the timing of these changes. In many real-world datasets [5,6,7], the ground truth is not explicitly known. However, we have a list of real-world events (i.e. MIT academic calendar and public news on Enron) that can serve as reference. There is no widely accepted benchmark that outlines which real-world events should be considered as true changes, particularly in the context of dynamic graphs. Hence, in our study, we validate the proposed method on real data, by identifying significant real-world events that align with our detected change points and by verifying with the competitor methods.
>
> [1] Shen, C., Larson, J., Trinh, H., Qin, X., Park, Y., \& Priebe, C. E.,``Discovering Communication Pattern Shifts in Large-Scale Networks using Encoder Embedding and Vertex Dynamics", IEEE TNSE, 2024.
>
> [2] Gallagher, I., Jones, A., \& Rubin-Delanchy, P. ,``Spectral embedding for dynamic networks with stability guarantees", NeurIPS, 2021.
>
>
> [3] Boyd, S., Parikh, N., Chu, E., Peleato, B., \& Eckstein, J. ,``Distributed Optimization and Statistical Learning via the Alternating Direction Method of Multipliers", Foundations and Trends in Machine learning, 2011.
>
> [4] Wang, Y., Yin, W., \& Zeng, J.,``Global Convergence of ADMM in Nonconvex Nonsmooth Optimization", Journal of Scientific Computing, 2019.
>
> [5] Madrid Padilla, O. H., Yu, Y., \& Priebe, C. E., ``Change Point Localization in Dependent Dynamic Nonparametric Random Dot Product Graphs", Journal of Machine Learning Research, 2022.
>
> [6] Chen, T., Lubberts, Z., Athreya, A., Park, Y., \& Priebe, C. E., ``Euclidean Mirrors and First-order Changepoints in Network Time Series", arXiv preprint, 2024.
>
> [7] Van den Burg, G. J., \& Williams, C. K., ``An Evaluation of Change Point Detection Algorithms", arXiv preprint, 2020.

---

### Review · Reviewer_97fx · 2024-10-05

**Summary Of Contributions:**

The paper proposes a methodology for changepoint detection in dynamics graphs. The proposal is presented in a clear and elegant, yet simple, manner.

The proposed generative model operates using a compressed latent representation of the graphs, which is linked to the graph via a (decoder) neural net. The changepoints are detected via a group fused Lasso formulation solved via ADMM. The article then provides synthetic and real-world experiments, as well as a comparison with other benchmark models.

Overall, the topic addressed by the authors is very relevant, and their contribution to that topic is reasonable.

**Audience:**

Yes

**Claims And Evidence:**

No

**Requested Changes:**

1)
The paper is well written in general, but note that page 11: tables 2 - > table 2 (same with table 3)

2)
In the Enron experiment, besides the bottom row, the discrepancy between detected changes and possible trigger events ranges between 10 and 40 days. Due to my lack of familiarity with the dynamics of such events, I cannot confirm or reject the relationship between the detected changepoints and the claimed events. As a matter of fact, from Fig 6, it seems that CPDstergm was able to detect the event "2001-08-14 CEO resigned" better than the proposed method, though it is unclear from the figure due to the lack of labels in the x-axis.

Perhaps the authors can include a discussion in this regard to justify their detected changepoints.

**Strengths And Weaknesses:**

The main strengths of the manuscript, as per the comment above, are clarity, relevance, and timeliness. Due to my lack of familiarity with the problem addressed by the authors, I couldn't possibly assess the submission in terms of novelty.

Though they are presented clearly, the experimental validation can be strengthened. I acknowledge that evaluating changepoint detection methods is challenging due to the lack of ground truth, and I understand that in this case (graphs) relevant data can be even more scarce. In general, there is little confirmation of the ability of the proposed method to detect changes in real-world data: there is no quantitative assessment in Sec 5.2 and 5.3. This quantitative evaluation is needed, as the abstract reads " [Experiments in real data] demonstrate the ability of the generative model in supporting change point detection with good performance."

---

> ### Author Response · Authors · 2024-10-13
> **Response to Review Comments**
>
> We thank the reviewer for pointing out the justification of the detected change points in real-data experiments. We have also uploaded an updated manuscript based on the requested changes.
>
> $\bf{Q1}$: Thanks! We have corrected the typos.
>
> $\bf{Q2}$: We have modified the manuscript for this comment from Line 267 to Line 270 on Page 14. Below are our responses to this comment.
>
> As change point detection is often discussed in an unsupervised learning setting, there may not be a widely accepted ground truth for significant change points, particularly in the context of dynamic graphs. In our real-data experiments, we validated the proposed method by aligning the detected change points with notable real-world events and further confirmed our findings by comparing them to the results of competitor methods.
>
> After reviewing the Enron timeline, we found that an important employee meeting took place on 2001-09-26, during which the CEO assured employees that Enron stock was a good buy and that the company’s accounting methods were legal and appropriate. Following the week of 2001-09-24, Enron's stock surged briefly before its continued decline, leading to the company's bankruptcy in December 2001. We believe it is possible that this meeting, which falsely encouraged employees to buy stock, had a significant impact on the email communication patterns among the employees, aligning more closely with our detected change point of 2001-09-24.

---

### Decision · Action_Editor_63Hj · 2024-11-21

**Recommendation:** Reject

**Comment:**

The method is certainly interesting and potentially useful, as agreed by all reviewers. However, the authors have not made a strong case why their change points are "better" than those found by competing methods (see "Claims and Evidence"). I would like to see a resubmission with more thorough experimental evaluation that includes an explanation of the differences between the methods found.

Some additional points made by the reviewers in discussion:
* The complexity order should include the number of nodes as a parameter.
* It is not clear why node-level methods such as Larroca et al (2021, Marenco et al (2022) and Gong et al (2023) would not be powerful enough to capture information on the whole graph; it would be nice to see this evaluated experimentally.
* We suggest empirically validating the hypothesis that the low-rank decoder avoids overfitting, particularly on sparse graphs.
* The new version of the manuscript does not include an URL to the source code.

**Audience:**

Yes, this would be of interest to people interested in graph generative models and change point detection.

**Claims And Evidence:**

The synthetic experiments do a reasonable job of conveying the benefits of the approach; however the real-data experiments are less convincing. The reviewers point out that evaluating change-point detection is hard since it is inherently an unsupervised approach, which is of course true. However, it is hard to see why we would prefer the authors' method vs the competing methods, given figures 5 and 6. The authors provide qualitative assessments of why their change points are reasonable, but there is no discussion as to why the change points found by other methods might *not* be reasonable. I would like to see discussion of why change-points found by multiple other methods are not present in the authors results. For example, Kei et al (2023) provides justification for several change-points that are found by CPDstergm but not by the authors' method; similarly, Song and Chen (2022b) justify their change points which differ from those found by the author. Perhaps more qualitative assessment of the synthetic experiments would allow us to better understand failure modes of the alternative methods?

**Resubmission Of Major Revision:**

The authors may consider submitting a major revision at a later time.